# DUAL-ROBUST CROSS-DOMAIN OFFLINE REINFORCEMENT LEARNING AGAINST DYNAMICS SHIFTS

**Zhongjian Qiao**[1], **Rui Yang**[2], **Jiafei Lyu**[6], **Xiu Li**[3], **Zhongxiang Dai**[5], **Zhuoran Yang**[4], **Siyang Gao**[1], **Shuang Qiu**[1*]

[1]CityUHK, [2]UIUC, [3]Tsinghua University, [4]Yale University, [5]CUHK(SZ), [6]Tencent
`zhongqiao2-c@my.cityu.edu.hk, shuanqiu@cityu.edu.hk`

## ABSTRACT

Single-domain offline reinforcement learning (RL) often suffers from limited data coverage, while cross-domain offline RL handles this issue by leveraging additional data from other domains with dynamics shifts. However, existing studies primarily focus on train-time robustness (handling dynamics shifts from training data), neglecting the test-time robustness against dynamics perturbations when deployed in practical scenarios. In this paper, we investigate dual (both train-time and test-time) robustness against dynamics shifts in cross-domain offline RL. We first empirically show that the policy trained with cross-domain offline RL exhibits fragility under dynamics perturbations during evaluation, particularly when target domain data is limited. To address this, we introduce a novel robust cross-domain Bellman (RCB) operator, which enhances test-time robustness against dynamics perturbations while staying conservative to the out-of-distribution dynamics transitions, thus guaranteeing the train-time robustness. To further counteract potential value overestimation or underestimation caused by the RCB operator, we introduce two techniques, the dynamic value penalty and the Huber loss, into our framework, resulting in the practical **D**ual-**RO**bust **C**ross-domain **O**ffline RL (DROCO) algorithm. Extensive empirical results across various dynamics shift scenarios show that DROCO outperforms strong baselines and exhibits enhanced robustness to dynamics perturbations. Code is available at `https://github.com/zq2r/DROCO.git`.

## 1 INTRODUCTION

Reinforcement learning (RL) (Sutton & Barto, 1999) has been a vital tool in various fields, such as embodied manipulation (Zakka et al., 2023; Jiao et al., 2025) and natural language processing (Ouyang et al., 2022; Rafailov et al., 2023). The success of typical RL often relies on numerous online interactions with the environment, which can be costly or even risky in the real world. Offline RL (Levine et al., 2020), instead, trains the policy with only a pre-logged offline dataset, eliminating the need for interactions with the environment. However, offline RL often struggles with a limited offline dataset. To address this, recent studies (Wen et al., 2024; Lyu et al., 2025; Liu et al., 2022) have explored Cross-Domain Offline RL. In this setting, data from the target domain is limited, but we have access to datasets from a relevant but distinct domain (the source domain), which may contain sufficient offline data. The goal of cross-domain offline RL is to utilize the datasets from both the source domain and the target domain to learn an effective policy for the target environment.

Although cross-domain offline RL is promising, simply merging the source domain dataset and target domain dataset for policy training induces policy divergence and suboptimal performance (Wen et al., 2024). The issue stems from the dynamics mismatch: the transition dynamics of the source domain may differ from that of the target domain. Recent advances tackle this issue by learning domain classifiers to estimate the dynamics gap (Liu et al., 2022), or by filtering source domain data based on mutual information (Wen et al., 2024) or optimal transport (Lyu et al., 2025). These works focus on enhancing the *train-time robustness* of the policy against dynamics shifts, that is, handling the source-target dynamics mismatch. However, they overlook the occurrence of potential dynamics

---

*Corresponding Author

shifts during the deployment of the learned policy in real-world environments. For example, an RL policy for robotics manipulation is trained on data collected from a real robot (target domain data) and an imperfect simulator (source domain data). When the policy is deployed on the real robot, the robot's physical components may degrade over time, causing the transition dynamics to deviate from that observed in the target domain dataset. Consequently, the policy's performance may deteriorate during deployment, highlighting the need for methods that ensure *test-time robustness*, that is, addressing the dynamics mismatch between the target and deployment environment.

In this paper, we initiate the investigation of dual (both train-time and test-time) robustness to dynamics shifts in cross-domain offline RL. We first empirically show that with limited target domain data, the learned policy could be highly fragile to test-time dynamics shifts. To address this issue, we propose **D**ual-**RO**bust **C**ross-domain **O**ffline RL (DROCO), bringing a new perspective on robustness specifically tailored for cross-domain offline RL, going beyond single-domain robust RL (Iyengar, 2005; Kuang et al., 2022). The core component of DROCO is a novel robust cross-domain Bellman (RCB) operator, which we theoretically prove enhances test-time robustness against dynamics perturbations while remaining conservative to the out-of-distribution (OOD) dynamics transitions (Liu et al., 2024a), thus guaranteeing train-time robustness. However, value overestimation or underestimation may occur when using the RCB operator. To mitigate this, we introduce two techniques, the dynamic value penalty and the Huber loss (Huber, 1973), to our framework, resulting in our practical DROCO algorithm. Our contributions are summarized as follows.

- We empirically demonstrate the fragility of cross-domain offline RL to test-time dynamics shifts and initiate the study of dual robustness in this setting, contributing new perspectives to the field.
- We introduce a novel RCB operator that is theoretically proven to achieve dual robustness against dynamics shifts. We further introduce dynamic value penalty and Huber loss to mitigate value overestimation or underestimation, yielding our practical algorithm, DROCO.
- Extensive experiments across diverse dynamics shift scenarios including kinematic and morphology shifts demonstrate that DROCO outperforms strong baselines and exhibits significant robustness against various test-time dynamics perturbations.

## 2 PRELIMINARIES

**RL.** We consider a Markov Decision Process (MDP) (Puterman, 1990) which is defined by the six-tuple $\mathcal{M} = (\mathcal{S}, \mathcal{A}, P, r, \rho, \gamma)$ where $\mathcal{S}$ is the state space, $\mathcal{A}$ is the action space, $P : \mathcal{S} \times \mathcal{A} \to \Delta(\mathcal{S})$ is the transition dynamics, $\Delta(\cdot)$ is the probability simplex, $r(s, a) : \mathcal{S} \times \mathcal{A} \to [-r_{\max}, r_{\max}]$ is the reward function, $\rho$ is the initial state distribution, and $\gamma$ is the discount factor. The objective of RL is to learn a policy $\pi : \mathcal{S} \to \Delta(\mathcal{A})$ that maximizes the expected discounted cumulative return $\mathbb{E}_\pi \left[ \sum_{t=0}^{\infty} \gamma^t r(s_t, a_t) \right]$. We define $Q^\pi(s, a) := \mathbb{E}_\pi \left[ \sum_{t=0}^{\infty} \gamma^t r(s_t, a_t) | s_0 = s, a_0 = a \right]$ and $V^\pi(s) := \mathbb{E}_{a \sim \pi(\cdot|s)} [Q^\pi(s, a)]$.

**Cross-Domain RL.** In cross-domain RL, we have access to a *source domain* MDP $\mathcal{M}_{\text{src}} = (\mathcal{S}, \mathcal{A}, P_{\text{src}}, r, \rho, \gamma)$ and a *target domain* MDP $\mathcal{M}_{\text{tar}} = (\mathcal{S}, \mathcal{A}, P_{\text{tar}}, r, \rho, \gamma)$. The only difference between the two domains is the transition dynamics, as considered by previous works (Wen et al., 2024; Lyu et al., 2025; Qiao et al., 2025b). In the offline setting, only a target domain dataset $\mathcal{D}_{\text{tar}}$ and a source domain dataset $\mathcal{D}_{\text{src}}$ are available. We aim to leverage the mixed dataset $\mathcal{D}_{\text{src}} \cup \mathcal{D}_{\text{tar}}$ to learn a well-performing agent in the target domain.

**Enhancing Robustness in RL.** Robust RL aims to optimize the worst-case policy performance to enhance the robustness against environmental perturbations. Different from standard RL, robust RL applies the following *robust Bellman operator* for Bellman backup:

$$\mathcal{T}_{\text{robust}} Q(s, a) = r(s, a) + \gamma \inf_{\mathcal{M} \in \mathcal{M}_\epsilon} \mathbb{E}_{s' \sim P_{\mathcal{M}}(\cdot|s,a)} \left[ \max_{a' \in \mathcal{A}} Q(s', a') \right],$$

where $\mathcal{M}_\epsilon$ is the dynamics uncertainty set under some distance metric. If we choose Wasserstein distance (Villani et al., 2008) as the distance metric, then $\mathcal{M}_\epsilon$ is the Wasserstein uncertainty set:

$$\mathcal{M}_\epsilon = \{\widehat{\mathcal{M}} : \mathcal{W}\left(P_{\mathcal{M}}(\cdot|s,a), P_{\widehat{\mathcal{M}}}(\cdot|s,a)\right) \leq \epsilon\}, \tag{1}$$

where $\mathcal{W}\left(P_{\mathcal{M}}(\cdot|s,a), P_{\widehat{\mathcal{M}}}(\cdot|s,a)\right) = \inf_{\gamma \in \Gamma(P_{\mathcal{M}}, P_{\widehat{M}})} \mathbb{E}_{s_1', s_2' \sim \gamma} [d(s_1', s_2')]$ is the Wasserstein distance between $P_{\mathcal{M}}(\cdot|s,a)$ and $P_{\widehat{\mathcal{M}}}(\cdot|s,a)$, $\Gamma(\cdot, \cdot)$ is the joint distribution, and $d(\cdot, \cdot)$ is an element-wise distance metric such as the Euclidean distance.

## 3 IS CROSS-DOMAIN OFFLINE RL SENSITIVE TO TEST-TIME DYNAMICS PERTURBATIONS?

To motivate our approach, we conduct an empirical study on the sensitivity of cross-domain offline RL to test-time perturbations. Our key finding is that cross-domain offline RL could be *highly sensitive* to test-time dynamics perturbations, especially when limited target domain data is given. Therefore, enhancing test-time robustness is crucial for cross-domain offline RL.

We adopt the `hopper-v2` task from Mu-JoCo (Todorov et al., 2012) as our target domain, and the full-size `hopper-expert-v2` dataset from D4RL (Fu et al., 2020) as the target domain dataset. To simulate dynamics shifts in the source domain, we create a modified `hopper-v2` environment with *kinematic shifts* (called `hopper-kinematic-v2`) by constraining the robot's joint rotation range. For the source domain dataset, we train an expert-level SAC (Haarnoja et al., 2018) policy and collect 1M samples in `hopper-kinematic-v2` environment with it. To examine the test-time robustness of cross-domain offline RL, we first train a policy using IGDF (Wen et al., 2024) on the full-size source and target domain datasets for 1M steps. We then evaluate the trained policy under four

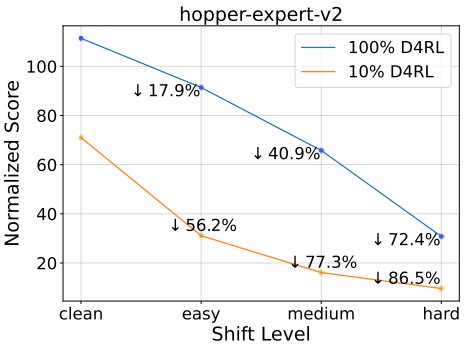

Figure 1: Performance comparison with different dataset sizes under dynamics perturbations.

conditions: (1) the original target environment (*clean*), and (2-4) kinematic perturbations with three levels (*easy*, *medium*, *hard*) following Lyu et al. (2024b). As shown by the blue curve in Figure 1, the policy demonstrates vulnerability to intense dynamics shifts, with performance degradation of 40.9% (*medium*) and 72.4% (*hard*) compared to the *clean* environment.

To better mimic the challenges when target domain data is limited in cross-domain offline RL, we construct a reduced target domain dataset by sampling only 10% of the `hopper-expert-v2` dataset. Our experiments reveal that the policy trained with this limited target data (while retaining full source domain data) is significantly more vulnerable to dynamics perturbations. As illustrated by the orange curve in Figure 1, performance degradation intensifies across all shift levels compared to the full-data case, demonstrating substantially reduced test-time robustness.

We attribute this phenomenon to the discrepancy between the true dynamics and the observed dynamics in the target domain dataset, whose magnitude inversely correlates with the dataset size. This discrepancy causes the policy to overfit to the dataset dynamics, thereby reducing its robustness to dynamics perturbations. These results highlight the necessity of enhancing test-time robustness against dynamics shifts for cross-domain offline RL, which we address in the following section.

## 4 DUAL-ROBUST CROSS-DOMAIN OFFLINE RL

In this section, we present dual-robust cross-domain offline RL. Inspired by Liu et al. (2024c), we first define the robust cross-domain Bellman (RCB) operator and additionally provide a practical version of it. We then show that dual-robustness can be achieved by applying the RCB operator solely on the source domain data. Finally, we present our practical algorithm, DROCO.

### 4.1 ROBUST CROSS-DOMAIN BELLMAN OPERATOR

**Definition 4.1** (RCB operator). *The robust cross-domain Bellman (RCB) operator $\mathcal{T}_{\mathrm{RCB}}$ is defined as*

$$\mathcal{T}_{\mathrm{RCB}}Q = \begin{cases} r + \gamma \mathbb{E}_{s' \sim P_{\mathcal{M}}} \left[ \max_{a' \sim \hat{\mu}(\cdot|s')} Q(s', a') \right], & \text{if } \mathcal{M} = \mathcal{M}_{\mathrm{tar}} \\ r + \gamma \inf_{\widehat{\mathcal{M}} \in \mathcal{M}_{\epsilon}} \mathbb{E}_{s' \sim P_{\widehat{\mathcal{M}}}} \left[ \max_{a' \sim \hat{\mu}(\cdot|s')} Q(s', a') \right], & \text{if } \mathcal{M} = \mathcal{M}_{\mathrm{src}}, \end{cases} \tag{2}$$

where $\hat{\mu}(\cdot|\cdot)$ is the behavior policy, and $\max_{a'\sim\hat{\mu}(\cdot|s')} Q(s',a')$ denotes taking maximum over actions in the support of $\hat{\mu}(\cdot|s')$, i.e., $\max_{a'\in\mathcal{A} \, s.t. \, \hat{\mu}(a'|s')>0} Q(s',a')$.

In Equation 2, we assume that the source and target domain datasets share the same behavior policy $\hat{\mu}$, following Wen et al. (2024). Note that this assumption is **only for notational simplicity.** Even if it does not hold, we could replace $\hat{\mu}$ with the respective behavior policies without affecting our analysis. The basic idea behind the RCB operator is that if $(s,a,s')$ comes from the target domain dataset, we use the standard in-sample Bellman operator (Kostrikov et al., 2021; Xu et al., 2023) for backup to enhance the performance; if the data are sampled from the source domain dataset, we apply the in-sample robust Bellman operator (which integrates in-sample learning into the robust Bellman operator) to achieve dual robustness to dynamics shifts, which we will discuss later.

We now characterize the dynamic programming property of the RCB operator and give the following proposition. All proofs are deferred to Appendix B.

**Proposition 4.1** ($\gamma$-contraction). *The RCB operator is a $\gamma$-contraction operator in the complete state-action space $(\mathbb{R}^{|\mathcal{S}\times\mathcal{A}|}, \|\cdot\|_\infty)$ where $\|\cdot\|_\infty$ denotes the $\ell_\infty$ norm, i.e., $\|\mathcal{T}_{RCB}Q_1 - \mathcal{T}_{RCB}Q_2\|_\infty \leq \gamma\|Q_1 - Q_2\|_\infty$ for any Q-functions $Q_1$ and $Q_2$.*

Proposition 4.1 presents that the RCB operator is a $\gamma$-contraction in the tabular MDP setting. However, directly applying the RCB operator for backup is unrealistic, since we are not available to the uncertainty set $\mathcal{M}_\epsilon$, given that the source environment is a black box. To handle this issue, we introduce the following dual reformulation of Equation 2 under the Wasserstein distance measure.

**Proposition 4.2** (Dual Reformulation). *Letting $\mathcal{M}_\epsilon$ be the Wasserstein uncertainty set defined by Equation 1, then the term $\inf_{\widehat{\mathcal{M}}\in\mathcal{M}_\epsilon} \mathbb{E}_{s'\sim P_{\widehat{\mathcal{M}}}} \left[ \max_{a'\sim\hat{\mu}(\cdot|s')} Q(s',a') \right]$ in Equation 2 is equivalent to*

$$\mathbb{E}_{s'\sim P_\mathcal{M}} \left[ \inf_{\bar{s}} \max_{a'\sim\hat{\mu}(\cdot|\bar{s})} Q(\bar{s},a') \right], \quad s.t. \quad d(s',\bar{s}) \leq \epsilon.$$

Proposition 4.2 provides a solution for transforming the intractable dynamics disturbance into the tractable state perturbations. Based on Proposition 4.2, we propose the practical RCB operator.

**Definition 4.2** (Practical RCB operator). *The practical RCB operator $\widehat{\mathcal{T}}_{\mathrm{RCB}}$ is defined as*

$$\widehat{\mathcal{T}}_{\mathrm{RCB}}Q = \begin{cases} r + \gamma\mathbb{E}_{s'\sim P_\mathcal{M}} \left[ \max_{a'\sim\hat{\mu}(\cdot|s')} Q(s',a') \right], & \text{if } \mathcal{M} = \mathcal{M}_{\mathrm{tar}} \\ r + \gamma\mathbb{E}_{s'\sim P_\mathcal{M}} \left[ \inf_{\bar{s}\in U_\epsilon(s')} \max_{a'\sim\hat{\mu}(\cdot|\bar{s})} Q(\bar{s},a') \right], & \text{if } \mathcal{M} = \mathcal{M}_{\mathrm{src}} \end{cases}$$

*where $U_\epsilon(s') = \{\bar{s}\in\mathcal{S} \mid d(s',\bar{s})\leq\epsilon\}$ is the state uncertainty set.*

The key distinction between $\widehat{\mathcal{T}}_{\mathrm{RCB}}$ and $\mathcal{T}_{\mathrm{RCB}}$ lies in their Bellman target computation for source domain data. While $\mathcal{T}_{\mathrm{RCB}}$ requires the dynamics uncertainty set $\mathcal{M}_\epsilon$ that is typically unavailable, $\widehat{\mathcal{T}}_{\mathrm{RCB}}$ solely relies on the state uncertainty set $U_\epsilon(s')$. Since $s'$ is observable in the source domain dataset, $U_\epsilon(s')$ can be constructed through noise perturbations of $s'$. This makes $\widehat{\mathcal{T}}_{\mathrm{RCB}}$ more feasible for Bellman backup than $\mathcal{T}_{\mathrm{RCB}}$, and the subsequent analyses are based on $\widehat{\mathcal{T}}_{\mathrm{RCB}}$. Moreover, the following proposition shows that $\widehat{\mathcal{T}}_{\mathrm{RCB}}$ still possesses the same favorable property as $\mathcal{T}_{\mathrm{RCB}}$, i.e., $\widehat{\mathcal{T}}_{\mathrm{RCB}}$ remains a $\gamma$-contraction.

**Proposition 4.3** ($\gamma$-contraction). *The practical RCB operator is a $\gamma$-contraction operator in the space $(\mathbb{R}^{|\mathcal{S}\times\mathcal{A}|}, \|\cdot\|_\infty)$, i.e., $\|\widehat{\mathcal{T}}_{RCB}Q_1 - \widehat{\mathcal{T}}_{RCB}Q_2\|_\infty \leq \gamma\|Q_1 - Q_2\|_\infty$ for any $Q_1$ and $Q_2$.*

## 4.2 DUAL ROBUSTNESS AGAINST DYNAMICS SHIFTS

In this section, we conduct a comprehensive analysis of both train-time and test-time robustness against dynamics shifts when employing the practical RCB operator. We first make the Lipschitz continuity assumption about the learned $Q$ function, which is widely used in prior theoretical studies of RL (Mao et al., 2024; Ran et al., 2023; Xiong et al., 2022; Liu et al., 2024b).

**Assumption 4.1** (Lipschitz $Q$ function). *The learned Q function is $K_Q$-Lipschitz w.r.t. state $s$, i.e., $\forall a\in\mathcal{A}, \forall s_1, s_2\in\mathcal{S}, |Q(s_1,a) - Q(s_2,a)| \leq K_Q\|s_1 - s_2\|$.*

We then analyze the train-time robustness against dynamics shifts from source domain data. Standard Bellman updates on source domain data might cause $Q$ overestimation due to OOD dynamics issues (Liu et al., 2024a; Niu et al., 2022), necessitating a conservative $Q$ estimation for robust performance. Proposition 4.4 shows that the learned $\widehat{Q}_{\text{RCB}}$ maintains bounded by applying $\widehat{\mathcal{T}}_{\text{RCB}}$.

**Proposition 4.4** (Train-time robustness against dynamics shifts). *Assume that $\epsilon$ is chosen such that* $\text{supp}\big(P_{\text{tar}}(\cdot \mid s, a)\big) \subseteq U_{\epsilon}(s'_{\text{src}})$ *for all* $(s, a, s'_{\text{src}}) \sim \mathcal{D}_{\text{src}}$. *Then under Assumption 4.1, the learned $Q$ function by applying $\widehat{\mathcal{T}}_{RCB}$ satisfies:*

$$Q^{\star}_{\hat{\mu}}(s, a) - \frac{2\gamma\epsilon K_Q}{1-\gamma} \leq \hat{Q}_{RCB}(s, a) \leq Q^{\star}_{\hat{\mu}}(s, a), \quad \forall (s, a) \in \mathcal{D}_{\text{src}},$$

*where $Q^{\star}_{\hat{\mu}}$ is the $Q$ function of optimal $\hat{\mu}$-supported policy[1] in the target domain.*

Proposition 4.4 suggests that, if a proper $\epsilon$ is chosen such that the uncertainty set $U_{\epsilon}(s'_{\text{src}})$ covers the support of $P_{\text{tar}}(\cdot|s, a)$, then the erroneous value overestimation will not occur, and the OOD dynamics issue is mitigated. Thus, the train-time robustness against dynamics shifts is guaranteed. We then analyze the test-time robustness against the environmental dynamics perturbations. Let $\pi_{\text{RCB}}$ and $\widehat{V}_{\text{RCB}}$ be the policy and value function learned by applying the practical RCB operator, respectively. When the target environment undergoes dynamics perturbations $(P_{\text{tar}}(\cdot) \to P_{\text{per}}(\cdot))$, the value function of $\pi_{\text{RCB}}$ within perturbed dynamics $P_{\text{per}}$, denoted as $V^{\pi_{\text{RCB}}}_{\text{per}}$, is bounded by Proposition 4.5.

**Proposition 4.5** (Test-time robustness against dynamics shifts). *Assume that $\epsilon$ is chosen such that* $\text{supp}\big(P_{\text{tar}}(\cdot \mid s, a)\big) \subseteq U_{\epsilon}(s'_{\text{src}})$ *for all* $(s, a, s'_{\text{src}}) \sim \mathcal{D}_{\text{src}}$. *As long as* $\mathcal{W}(P_{\text{per}}(\cdot|s, a), P_{\text{tar}}(\cdot|s, a)) \leq c$, *then for $\forall s_0 \in \mathcal{D}_{\text{src}}$, we have*

$$V^{\pi_{\text{RCB}}}_{\text{per}}(s_0) \geq \widehat{V}_{\text{RCB}}(s_0), \tag{3}$$

*where $c = \max \{c\,|\,U_c(s'_{\text{tar}}) \subseteq U_{\epsilon}(s'_{\text{src}}), s'_{\text{tar}} \sim P_{\text{tar}}(\cdot|s, a), (s, a, s'_{\text{src}}) \sim \mathcal{D}_{\text{src}}\}$.*

Proposition 4.5 gives that, for any disturbance with intensity below the threshold $c$ (measured in Wasserstein distance), the value of the learned policy $\pi_{\text{RCB}}$ in $P_{\text{per}}$ exceeds $\widehat{V}_{\text{RCB}}$. This implies that for any initial state $s_0 \in \mathcal{D}_{\text{src}}$, $\pi_{\text{RCB}}$ achieves better performance under perturbed dynamics $P_{\text{per}}$ than in the worst-case scenario, thereby improving test-time robustness against dynamics shifts. Furthermore, Proposition 4.4 and Proposition 4.5 reveal that, **(1)** dual robustness can be achieved by solely applying the RCB operator to the source domain data; **(2)** there is a trade-off between the two robustness notions, which is controlled by $\epsilon$. More discussions can be found in Appendix A.

## 4.3 PRACTICAL ALGORITHM

In Section 4.1, we formalize the practical RCB operator. Its application presents two key challenges: (1) determining the uncertainty set $U_{\epsilon}(s'_{\text{src}})$; (2) computing the minimum $Q$ value within this set. Although one can fix $\epsilon$ and adopt random sampling within $U_{\epsilon}(s'_{\text{src}})$, it lacks flexibility. In addition, if $P_{\text{src}}$ deviates far from $P_{\text{tar}}$, then $\epsilon$ would be too large, leading to overconservatism, compromising the performance. To address these limitations, we propose our practical algorithm, DROCO.

**Determining the uncertainty set via ensemble dynamics modeling.** Instead of fixing $\epsilon$ and randomly sampling from $U_{\epsilon}(s'_{\text{src}})$, DROCO first trains an ensemble dynamics model (Janner et al., 2019; Yu et al., 2020; Liu et al., 2024c) $\widehat{P}_{\psi}(\cdot) = \{\widehat{P}_{\psi_i}(\cdot)\}_{i=1}^{N}$ on $\mathcal{D}_{\text{tar}}$ via maximum likelihood estimation (MLE) to simulate $P_{\text{tar}}(\cdot)$:

$$\mathcal{L}_{\psi_i} = \mathbb{E}_{(s, a, s') \in \mathcal{D}_{\text{tar}}} \left[ \log \widehat{P}_{\psi_i}(s'|s, a) \right], \qquad i = 1, 2, ..., N \tag{4}$$

then we use the ensemble prediction set $\mathcal{X} = \left\{ s'_1, \cdots, s'_N | s'_i \sim \widehat{P}_{\psi_i}(\cdot|s, a), (s, a) \in \mathcal{D}_{\text{src}} \right\}$ to approximate sampling from the uncertainty set. This replacement is motivated by two key insights: (1) dual robustness only requires the uncertainty set around support of $P_{\text{tar}}(\cdot|s, a)$ rather than $s'_{\text{src}}$, thus alleviating the unnecessary conservatism; (2) each ensemble member's prediction naturally serves as a sample from this uncertainty set. In this way, the practical RCB operator for source domain data becomes:

$$\widehat{\mathcal{T}}_{\text{RCB}} Q = r + \gamma \inf_{\{s'_i\}^N \sim \widehat{P}_{\psi_i}} \left[ \max_{a'_i \sim \hat{\mu}(\cdot|s'_i)} Q(s'_i, a'_i) \right], \qquad \text{if } \mathcal{M} = \mathcal{M}_{\text{src}}. \tag{5}$$

---

[1] $\pi(\cdot|s)$ is $\hat{\mu}$-supported if $\pi(a|s) = 0$ for any action $a$ that $\hat{\mu}(a|s) = 0$.

However, the ensemble prediction set cannot cover the support of $P_{\text{tar}}(\cdot)$ as required in Proposition 4.4, such that the overestimation of $Q$ value may still occur. Proposition 4.6 reveals that only limited overestimation would occur when applying Equation 5 as the Bellman target.

**Proposition 4.6** (Limited overestimation). *If* $\sup_{s,a} D_{TV}(\widehat{P}_\psi(\cdot|s,a), P_{\text{tar}}(\cdot|s,a)) \leq \epsilon < \frac{1}{2}$, *we have*

$$
\inf_{\{s'_i\}^N \sim \widehat{P}_{\psi_i}(\cdot|s,a)} \left[ \max_{a'_i \sim \hat{\mu}(\cdot|s'_i)} Q(s'_i, a'_i) \right] \leq \mathbb{E}_{s' \sim P_{\text{tar}}(\cdot|s,a)} \left[ \max_{a' \sim \hat{\mu}(\cdot|s')} Q(s', a') \right] + (1 - (1-2\epsilon)^N) \frac{r_{\max}}{1-\gamma}.
$$

Proposition 4.6 holds under the assumption that the prediction error of the dynamics model stays small, which is difficult to fulfill given that the target domain data is limited and the dynamics model tends to overfit. Moreover, value underestimation may also occur due to the infimum operator. Therefore, we introduce the following techniques for the underlying value estimation issue.

**Tackling overestimation and underestimation.** We adopt two techniques to address the value estimation issue: dynamic value penalty, and using Huber loss (Huber, 1973) for Bellman update.

Instead of directly using Equation 5 for Bellman backup, we introduce a value penalty term:

$$
u(s, a, s') = \mathbb{I}(s' \sim P_{\text{src}}(\cdot|s,a)) \cdot \left( \max_{a' \sim \hat{\mu}(\cdot|s')} Q(s', a') - \inf_{\{s'_i\}^N \sim \widehat{P}_{\psi_i}(\cdot|s,a)} \left[ \max_{a'_i \sim \hat{\mu}(\cdot|s'_i)} Q(s'_i, a'_i) \right] \right). \tag{6}
$$

We then unify the source and target dynamics in the practical RCB operator, reformulating it as

$$
\widehat{\mathcal{T}}_{\text{RCB}} Q(s, a) = r(s, a) + \gamma \mathbb{E}_{s' \sim P_{\mathcal{M}}(\cdot|s,a)} \left[ \max_{a' \sim \hat{\mu}(\cdot|s')} Q(s', a') - \beta \cdot u(s, a, s') \right], \tag{7}
$$

where $\mathcal{M} = \mathcal{M}_{\text{tar}}$ or $\mathcal{M}_{\text{src}}$ and $\beta$ serves as a dynamic penalty coefficient that provides flexible control over value estimation. Specifically, we recover the practical RCB operator by setting $\beta$ to 1.0, $\beta > 1.0$ will increase the penalty to mitigate value overestimation, and $\beta < 1.0$ reduces the penalty to alleviate value underestimation. Although the dynamics model and value penalty are widely applied in offline RL (Yu et al., 2020; Sun et al., 2023; Liu et al., 2024c; Qiao et al., 2025a), our difference lies in the specific usage of the dynamics model and design of the penalty term.

**Remark.** If we use IQL (Kostrikov et al., 2021) for policy optimization, then $\max_{a' \sim \hat{\mu}(\cdot|s')} Q(s', a') \approx V(s')$, and Equation 6 can be re-written as

$$
u(s, a, s') = \mathbb{I}(s' \sim P_{\text{src}}(\cdot|s,a)) \cdot \left( V(s') - \inf_{\{s'_i\}^N \sim \widehat{P}_{\psi_i}(\cdot|s,a)} [V(s'_i)] \right). \tag{8}
$$

We note that Equation 8 resembles the value discrepancy term in VGDF (Xu et al., 2024), which is $V(s') - \mathbb{E}_{\{s'_i\}^N \sim \widehat{P}_{\psi_i}(\cdot|s,a)} [V(s'_i)]$. However, we extend this term by incorporating additional penalties for test-time dynamics shifts, whereas VGDF only addresses train-time dynamics shifts.

The second technique we adopt is the Huber loss (Huber, 1973), a well-established technique for noise-resistant optimization (Yang et al., 2024b; Roy et al., 2021). We replace the regular $\ell_2$ loss in the Bellman update with the Huber loss:

$$
\mathcal{L}_Q = \mathbb{E}_{\mathcal{D}_{\text{src}}} \left[ l_\delta \left( Q(s, a) - \widehat{\mathcal{T}}_{\text{RCB}} Q(s, a) \right) \right] + \frac{1}{2} \mathbb{E}_{\mathcal{D}_{\text{tar}}} \left[ (Q(s, a) - \mathcal{T} Q(s, a))^2 \right], \tag{9}
$$

where $l_\delta(a) = \begin{cases} 0.5a^2, & |a| < \delta \\ \delta(|a| - 0.5\delta), & |a| \geq \delta \end{cases}$ with $\delta$ being the transition threshold and $\mathcal{T}$ being the standard Bellman operator for target domain data. Specifically, if a severe value estimation error occurs such that $|Q(s, a) - \widehat{\mathcal{T}}_{\text{RCB}} Q(s, a)| > \delta$, the $\ell_2$ loss would transition to $\ell_1$ loss to improve robustness against outliers. This technique helps mitigate value estimation error. The last step is to utilize offline RL algorithms such as IQL to optimize the policy as in other works (Lyu et al., 2025; Wen et al., 2024). We present the detailed pseudo-code of DROCO in Appendix D.2.

Table 1: **Evaluation Results with train-time kinematic shifts.** half=halfcheetah, hopp=hopper, walk=walker2d, m=medium, me=medium-expert, mr=medium-replay, e=expert. We report the normalized score evaluated in the target domain, and $\pm$ captures the standard deviation across 5 seeds.

| Dataset | IQL$^\star$ | CQL$^\star$ | BOSA | DARA | IGDF | OTDF | DROCO (Ours) |
|---------|------|------|------|------|------|------|--------------|
| half-m | **45.2** | 37.7 | 39.6 | 44.1 | **45.2**±0.1 | 42.2±0.1 | **45.3**±**0.2** |
| half-mr | 22.1 | 23.6 | **26.3** | 21.6 | 22.9±1.4 | 15.6±3.1 | **26.9**±**3.2** |
| half-me | 43.7 | 54.8 | 42.2 | 52.7 | 57.1±8.9 | 46.7±4.4 | **60.1**±**7.1** |
| half-e | 49.7 | 36.0 | **84.3** | 47.4 | 47.6±2.1 | 79.6±3.0 | 67.4±5.8 |
| hopp-m | 48.8 | 35.7 | **71.4** | 48.8 | 54.3±6.6 | 46.3±3.7 | 55.4±5.3 |
| hopp-mr | 40.2 | 43.2 | 29.5 | 41.6 | 30.0±5.2 | 26.2±4.4 | **47.3**±**7.0** |
| hopp-me | 12.5 | 7.8 | 49.6 | 17.0 | 11.6±0.6 | **58.1**±**4.9** | 54.0±6.4 |
| hopp-e | 62.6 | 47.9 | 94.8 | 59.1 | 70.1±3.2 | **97.0**±**3.3** | 89.3±9.6 |
| walk-m | 48.7 | 47.7 | 44.5 | 43.4 | 51.8±2.4 | 43.0±2.1 | **70.8**±**3.3** |
| walk-mr | 12.6 | 17.8 | 4.8 | 15.6 | 11.2±1.1 | 10.7±1.9 | **27.7**±**3.0** |
| walk-me | **95.4** | 61.4 | 35.1 | 85.3 | 90.6±3.4 | 63.1±6.6 | 78.5±6.7 |
| walk-e | 90.1 | 83.8 | 41.9 | 85.5 | 93.7±5.8 | 98.9±2.1 | **106.0**±**0.8** |
| ant-m | 89.9 | 58.2 | 28.4 | **98.9** | 88.0±4.6 | 86.1±3.7 | 92.7±6.3 |
| ant-mr | 46.8 | 39.4 | 22.0 | 42.1 | **58.2**±**9.7** | 39.6±8.1 | 44.8±4.5 |
| ant-me | 106.1 | 100.6 | 102.5 | 104.8 | 112.8±4.0 | 105.1±3.9 | **119.0**±**3.6** |
| ant-e | 111.0 | 94.3 | 57.6 | 115.1 | **119.2**±**5.6** | 111.6±2.9 | **120.0**±**2.1** |
| **Total** | 925.4 | 789.9 | 774.5 | 923.0 | 964.3 | 969.8 | **1105.2** |

## 5 EXPERIMENTS

In this section, we conduct extensive experiments to examine our method. We aim to answer the following two questions: (1) Can DROCO outperform prior strong baselines across various train-time dynamics shifts and dataset qualities? (2) Can DROCO show enhanced robustness against test-time dynamics perturbations? We also test the parameter sensitivity of DROCO.

### 5.1 MAIN RESULTS

**Experimental Settings.** Following previous works (Lyu et al., 2025; Wen et al., 2024), We employ 4 MuJoCo (Todorov et al., 2012) tasks as source domains: `halfcheetah-v2`, `hopper-v2`, `walker2d-v2`, and `ant-v3`. For the target domain datasets, we utilize 4 data qualities from D4RL (Fu et al., 2020) for each task: `medium`, `medium-replay`, `medium-expert`, and `expert`, totaling 16 target domain datasets. For the source domain, we introduce kinematic shifts and morphology shifts in the target domain setup. We collect source domain datasets with 4 data qualities, resulting in a total of 32 ($4[\texttt{tasks}] \times 2[\texttt{shift types}] \times 4[\texttt{data qualities}]$) source domain datasets. Each pair of the source and target domain datasets shares the same task type (such as `hopper-v2`) and dataset quality (such as `expert`). More details about the tasks and datasets can be found in Appendix C.1.

**Baselines.** We consider the following baselines: **IQL$^\star$**, **CQL$^\star$** (which train IQL (Kostrikov et al., 2021) and CQL (Kumar et al., 2020) with the mixed dataset $\mathcal{D}_{\text{tar}} \cup \mathcal{D}_{\text{src}}$), **BOSA** (Liu et al., 2024a), **DARA** (Liu et al., 2022), **IGDF** (Wen et al., 2024) and **OTDF** (Lyu et al., 2025).

**Results.** We run each baseline and DROCO for 1M training steps over 5 random seeds, and report the results with train-time kinematic shifts in Table 1 (the results with morphology shifts are deferred to Appendix E.2). Note that our evaluation is under the clean target environment. Empirical results demonstrate that DROCO achieves superior performance in **9** out of 16 tasks, outperforming all 6 baselines. Furthermore, in terms of the total normalized score, DROCO achieves a remarkable **1105.2**, significantly surpassing the second-best method, OTDF (969.8), by **14.0%**. We attribute the suboptimal performance of DROCO on the remaining datasets to its trade-off between performance and robustness, whereas other methods only consider performance. However, DROCO still achieves a competitive performance compared to other baselines on these datasets. These results indicate that DROCO exhibits superior train-time robustness against dynamics shifts.

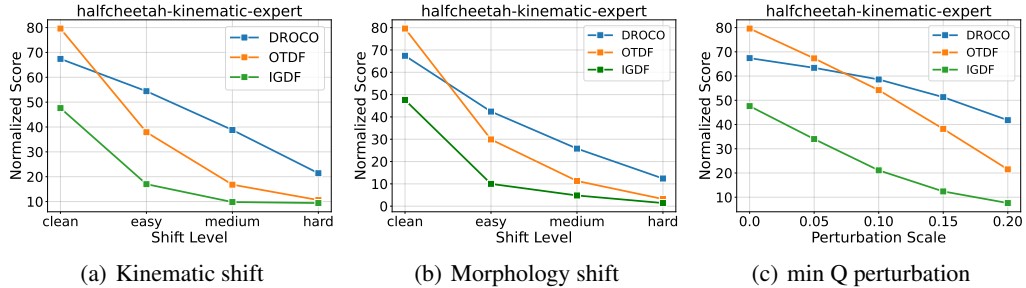

(a) Kinematic shift  (b) Morphology shift  (c) min Q perturbation

Figure 2: Evaluation results under different types and levels of dynamics perturbations.

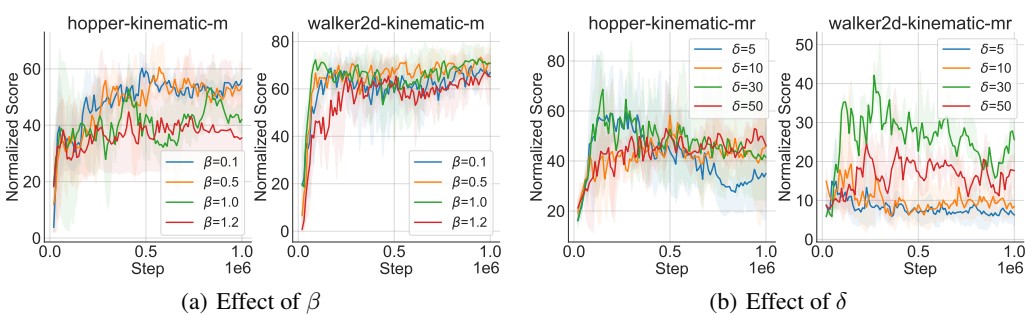

(a) Effect of $\beta$  (b) Effect of $\delta$

Figure 3: Parameter sensitivity experiments on $\beta$ and $\delta$.

## 5.2 EVALUATION UNDER DYNAMICS PERTURBATIONS

**Experimental Settings.** Test-time robustness can be measured by the degree of performance degradation in the face of dynamic perturbations, compared to the clean environment. To examine the test-time robustness of DROCO, we introduce three kinds of dynamics perturbations during evaluation in the target environment: kinematic perturbations, morphology perturbations, min Q perturbations (Yang et al., 2022; 2024b). The first two perturbation types mirror the source domain dynamics shifts, each implemented at three intensity levels following (Lyu et al., 2024b): *easy*, *medium*, and *hard*. The third perturbation type represents an adversarial attack strategy that modifies dynamics by finding the state $\bar{s}$ from $U_\epsilon(s')$ that minimizes the $Q$ value, i.e., $\bar{s} = \arg\min_{\bar{s} \in U_\epsilon(s')} Q(\bar{s}, \pi(\bar{s}))$, where $\epsilon$ is related to the perturbation scale. We consider DROCO to exhibit better test-time robustness if it demonstrates less performance degradation than the baselines under the same shift severity.

**Results.** We evaluate our method against two baselines (IGDF and OTDF) under three perturbation types with varying intensity levels. Due to space constraints, we only report results when the source domain dataset is `halfcheetah-kinematic-expert`, as illustrated in Figure 2. A wider range of evaluation can be found in Appendix E.3.

Our experiments demonstrate that DROCO exhibits superior robustness to dynamic perturbations compared to baseline methods. Specifically, under easy-level kinematic shifts, DROCO shows only a **19.3%** performance degradation (from 67.4 to 54.4), whereas both IGDF and OTDF suffer over $50\%$ performance deterioration. We notice that DROCO displays greater sensitivity to morphological perturbations than to kinematic perturbations, with a $42.1\%$ performance decrease under easy-level morphological variations. We attribute this to the absence of morphology shifts in the source domain data, rendering the policy less adaptable to this unseen perturbation type. Nevertheless, DROCO still outperforms both baselines: under the same conditions, OTDF and IGDF exhibit performance declines of $62.4\%$ and $78.9\%$ respectively. Notably, DROCO maintains consistent robustness against min Q perturbations across all scales. At the highest perturbation scale of 0.2, DROCO's performance decreases by **37.9%**, compared to $73.6\%$ and $84.0\%$ for OTDF and IGDF.

## 5.3 PARAMETER SENSITIVITY

We examine the sensitivity of DROCO to the introduced hyperparameters. There are two main hyperparameters in DROCO: the penalty coefficient $\beta$ and the transition threshold $\delta$.

**Penalty Coefficient $\beta$.** The parameter $\beta$ controls the intensity of value penalty. A larger $\beta$ leads to a stronger penalty and suppresses value overestimation, and vice versa. We sweep $\beta$ across $\{0.1, 0.5, 1.0, 1.2\}$ and show the experimental results with `medium` datasets in Figure 3 (a). We observe that different tasks prefer distinct $\beta$. For example, setting $\beta = 0.1$ achieves the best performance for `hopper-kinematic-medium`, while `walker2d-kinematic-medium` prefers $\beta = 1.0$.

**Transition Threshold $\delta$.** The parameter $\delta$ determines when the $\ell_2$ loss turns to $\ell_1$ loss for Bellman update. A larger $\delta$ corresponds to a more lenient transition condition. To test the effect of $\delta$, we select $\delta$ among $\{5, 10, 30, 50\}$ and conduct experiments with `medium-replay` datasets. The results in Figure 3 (b) indicate that a too small $\delta$ (e.g., $\delta = 5$) leads to inferior performance, while setting $\delta = 30$ achieves a good performance. However, we find the optimal $\delta$ varies across different tasks through additional experiments, and more discussions are provided in Appendix E.5.

**Remark.** Although different values of $\beta$ and $\delta$ are preferred for different tasks (as shown in Appendix E.5), we could still find some patterns across different tasks. We find that setting $\beta \leq 1.0$ works for most tasks, implying that value underestimation occurs more often due to the infimum operator. We also find that a larger $\delta$ (30 and 50) is preferred for most tasks. We believe it is because the $\ell_2$ loss is beneficial for training stability. Therefore, for a new task, we could first try $\beta \leq 1.0$ and $\delta = 30$ (or $\delta = 50$). This could serve as a guideline for finding the best hyperparameter.

## 6 RELATED WORK

**Offline RL.** In offline RL, a fixed dataset is provided, and no further interactions are allowed. As a result, conventional off-policy RL algorithms suffer from the extrapolation error due to OOD actions and exhibit poor performance (Kumar et al., 2020; Fujimoto et al., 2019). To address this challenge, various offline RL algorithms have been developed. Common solutions include incorporating policy constraints (Kumar et al., 2019; Fujimoto & Gu, 2021), learning a conservative value function (Kumar et al., 2020; Lyu et al., 2022; Jin et al., 2021; Zhang et al., 2024), leveraging a dynamics model to facilitate policy learning (Yu et al., 2020; 2021; Qiao et al., 2025a; 2024; 2026), performing in-sample learning to avoid querying OOD actions (Kostrikov et al., 2021; Xu et al., 2023; Garg et al., 2023; Zhang et al., 2023), etc. However, these methods require that the offline dataset contains a large amount of data. In contrast, we focus on cross-domain offline RL, which relaxes the target data coverage requirement.

**Cross-Domain RL.** Cross-domain RL (Niu et al., 2024) faces the challenge of domain mismatch, including observation mismatch (Yang et al., 2023), viewpoints mismatch (Liu et al., 2018; Sadeghi et al., 2018), and dynamics mismatch (Wen et al., 2024; Lyu et al., 2025; Xu et al., 2024; Niu et al., 2022; 2023), etc. In this paper, we exclusively focus on the dynamics mismatch. Previous studies handle this issue by adaptively penalizing $Q$ value on source domain samples (Niu et al., 2022), capturing dynamics mismatch from a representation learning perspective (Lyu et al., 2024a) and value discrepancy perspective (Xu et al., 2024), modifying the reward function in the source domain (Liu et al., 2022; Eysenbach et al., 2020; Xue et al., 2023; Wang et al., 2024), etc. We focus on the offline setting, where the current works (Liu et al., 2022; 2024a; Wen et al., 2024; Lyu et al., 2025; Wang et al., 2024; Yan et al., 2026) primarily consider the dynamics shifts from the source domain data, while we further consider the dynamics shifts from environmental perturbations.

**Robust RL.** Robust RL (Iyengar, 2005; Xu & Mannor, 2010) aims to learn a policy resilient to environmental perturbations or data corruption. One line of research in robust RL focuses on train-time robustness against data corruption (Yang et al., 2024b; Zhang et al., 2021; 2022; Ye et al., 2023; Yang et al., 2024a; Xu et al., 2025), while another line addresses test-time robustness against environmental perturbations (Yang et al., 2022; Zhihe & Xu, 2023; Shi & Chi, 2024; Liu et al., 2024c). These works focus only on a single perspective of robustness (train-time or test-time) and the single-domain offline settings. For the cross-domain setting, (He et al., 2025; Liu & Xu, 2024a; Van et al., 2025) study robust cross-domain RL, but they are different from our work, since they still only

consider one aspect of robustness and focus on the online setting. In contrast, our work addresses the cross-domain offline setting and jointly considers both train-time and test-time robustness.

# 7 CONCLUSION

In this paper, we investigate the dual (train-time and test-time) robustness against dynamics shifts in cross-domain offline RL. We propose a novel RCB operator and theoretically demonstrate its ability of dual robustness. To further handle the potential value estimation error, we add a dynamic value penalty and use Huber loss for Bellman update, yielding our practical DROCO algorithm. Through extensive experiments across various dynamics shift scenarios, we show that DROCO outperforms prior strong baselines and exhibits strong robustness to dynamics perturbations.

# 8 ACKNOWLEDGEMENTS

This research was supported in part by the Hong Kong Research Grants Council (GRF 11217925, GRF 16209124) and the National Science Foundation of China (Grant 72371214). The authors would also like to thank the anonymous reviewers for their valuable comments on our manuscript.

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

## A   More Discussions of DROCO

In this section, we provide further clarifications on several questions regarding DROCO that readers might be concerned about.

**1. Why not test DROCO on more tasks such as Antmaze and Adroit?**

In our experiments, we evaluate our method DROCO and other baselines on MuJoCo-based tasks (e.g., `halfcheetah-v2` and `hopper-v2`). This experimental setting is standard and has been adopted by recent works such as VGDF (Xu et al., 2024), IGDF (Wen et al., 2024), OTDF (Lyu et al., 2025), and CompFlow (Kong et al., 2025). By following these established settings, we believe our experiments are sufficient for evaluating DROCO's effectiveness.

On the other hand, existing literature (Lyu et al., 2024b) indicates that Antmaze tasks with varying map structures are highly challenging for cross-domain RL (Observation 3, p.7), as adapting policies across structural barriers remains difficult. Similarly, cross-domain RL methods often fail on dexterous hand manipulation tasks (Adroit) with kinematic or morphology shifts (Observation 4, p.8). Empirically, we find that not only DROCO but all baseline methods (e.g., BOSA, DARA, IGDF, OTDF) struggle to achieve meaningful performance on Antmaze and Adroit tasks.

We emphasize that enabling cross-domain RL to succeed in such challenging settings (Antmaze and Adroit) remains an open problem (p.17 in (Lyu et al., 2024b)) and falls beyond the scope of this work. We believe that MuJoCo tasks with dynamics shift provide a sufficient and appropriate testbed for evaluating our method.

**2. Can DROCO be extended into settings where the source and target domains have distinct state-action representations?**

The answer is yes. Although this work follows the setting of recent studies (e.g., BOSA, DARA, IGDF, OTDF) and assumes identical state and action spaces across source and target domains, DROCO can be generalized to domains with distinct state-action representations. This can be achieved by incorporating techniques such as inter-domain mapping via dynamics cycle consistency (Zhang et al., 2020). Other mapping methods (You et al., 2022; Pan et al.) are also compatible with our framework and could be seamlessly integrated. Thus, DROCO remains applicable even under varying state-action spaces.

**3. Why is the Wasserstein uncertainty set chosen instead of other uncertainty sets?**

Although there are other possible choices of uncertainty set like the $(s, a)$-rectangularity (Iyengar, 2005) and $d$-rectangularity (Liu & Xu, 2024b), their dual reformulations often result in complex constraints or regularizations and are typically limited to simple linear MDP settings. In contrast, the Wasserstein uncertainty set admits an elegant closed-form dual reformulation (Proposition 4.2), which allows converting dynamics perturbations into a simple state uncertainty set—a property critical for practical implementation. Moreover, the Wasserstein metric inherently couples state transitions, enabling a natural mapping to state perturbations and providing geometric interpretability.

**4. Why not apply the RCB operator on target domain data to improve test-time robustness?**

On the one hand, Propositions 4.4 and 4.5 show that dual robustness is guaranteed as long as $\mathrm{supp}(P_{\mathrm{tar}}(\cdot|s, a)) \subseteq U_\epsilon(s'_{\mathrm{src}})$ holds—even when the RCB operator is applied only to source domain data. Even when this condition is not fully satisfied, our techniques (dynamic value penalty and Huber loss) still enhance robustness. Therefore, applying the RCB operator to target domain data is unnecessary.

On the other hand, target domain data is important for achieving high performance in the clean target environment. Applying the RCB operator to it would introduce conservatism and compromise performance. Moreover, since the target data is scarce, any improvement in test-time robustness from using them would be limited. Thus, the optimal strategy is to apply the standard Bellman operator to target data to improve performance, and the RCB operator to source data to enhance robustness.

**5. Is there a trade-off between train-time and test-time robustness?**

There is a trade-off between train-time and test-time robustness, and it is controlled by $\epsilon$. Specifically, when $\mathrm{supp}(P_{\mathrm{tar}}(\cdot|s, a)) \subseteq U_\epsilon(s'_{\mathrm{src}})$ is satisfied, further increasing $\epsilon$ might bring excessive

conservatism. While this enhances test-time robustness against dynamics shifts (since $c$ is monotonically increasing with respect to $\epsilon$), it sacrifices performance on the clean target domain, thereby reducing train-time robustness.

# B   PROOFS OF PROPOSITIONS

## B.1   PROOF OF PROPOSITION 4.1

*Proof.* We recall the definition of the RCB operator below:

$$\mathcal{T}_{\text{RCB}}Q(s,a) = \begin{cases} r + \gamma \mathbb{E}_{s' \sim P_{\mathcal{M}}} \left[ \max_{a' \sim \hat{\mu}(\cdot|s')} Q(s',a') \right], & \text{if } \mathcal{M} = \mathcal{M}_{\text{tar}} \\ r + \gamma \inf_{\mathcal{M} \in \mathcal{M}_\epsilon} \mathbb{E}_{s' \sim P_{\mathcal{M}}} \left[ \max_{a' \sim \hat{\mu}(\cdot|s')} Q(s',a') \right], & \text{if } \mathcal{M} = \mathcal{M}_{\text{src}} \end{cases}.$$

Let $Q_1$ and $Q_2$ be two arbitrary $Q$ functions. Then for any state-action pair $(s,a)$, if the next state $s' \sim P_{\text{tar}}(\cdot|s,a)$, we have

$$\|\mathcal{T}_{\text{RCB}}Q_1 - \mathcal{T}_{\text{RCB}}Q_2\|_\infty = \gamma \max_{s,a} \left| \mathbb{E}_{s'} \left[ \max_{a' \sim \hat{\mu}(\cdot|s')} Q_1(s',a') - \max_{a' \sim \hat{\mu}(\cdot|s')} Q_2(s',a') \right] \right|$$

$$\le \gamma \max_{s,a} \mathbb{E}_{s'} \left| \max_{a' \sim \hat{\mu}(\cdot|s')} Q_1(s',a') - \max_{a' \sim \hat{\mu}(\cdot|s')} Q_2(s',a') \right|$$

$$\le \gamma \max_{s,a} \|Q_1 - Q_2\|_\infty$$

$$= \gamma \|Q_1 - Q_2\|_\infty,$$

where the second inequality holds from the fact that for any function $f_1$, $f_2$, any variant $x \sim \mathcal{X}$,

$$\left| \max_{x \sim \mathcal{X}} f_1(x) - \max_{x \sim \mathcal{X}} f_2(x) \right| \le \max_{x \sim \mathcal{X}} |f_1(x) - f_2(x)|. \tag{10}$$

If the next state $s' \sim P_{\text{src}}(\cdot|s,a)$, we have

$$\|\mathcal{T}_{\text{RCB}}Q_1 - \mathcal{T}_{\text{RCB}}Q_2\|_\infty$$

$$= \gamma \max_{s,a} \left| \inf_{\mathcal{M} \in \mathcal{M}_\epsilon} \mathbb{E}_{s'} \left[ \max_{a' \sim \hat{\mu}(\cdot|s')} Q_1(s',a') \right] - \inf_{\mathcal{M} \in \mathcal{M}_\epsilon} \mathbb{E}_{s'} \left[ \max_{a' \sim \hat{\mu}(\cdot|s')} Q_2(s',a') \right] \right|$$

$$\le \gamma \max_{s,a,s'} \left| \max_{a' \sim \hat{\mu}(\cdot|s')} Q_1(s',a') - \max_{a' \sim \hat{\mu}(\cdot|s')} Q_2(s',a') \right|$$

$$\le \gamma \max_{s,a} \|Q_1 - Q_2\|_\infty$$

$$= \gamma \|Q_1 - Q_2\|_\infty,$$

where the first inequality comes from the fact that for any function $f_1$, $f_2$, any variant $x \sim \mathcal{X}$,

$$\left| \min_{x \sim \mathcal{X}} f_1(x) - \min_{x \sim \mathcal{X}} f_2(x) \right| \le \max_{x \sim \mathcal{X}} |f_1(x) - f_2(x)|. \tag{11}$$

Combining the results, we conclude that the RCB operator is a $\gamma$-contraction operator in the complete state-action space, which naturally leads to the conclusion that any initial $Q$ function would converge to a unique fixed point by repeatedly applying $\mathcal{T}_{\text{RCB}}$. This completes the proof. $\qquad \square$

## B.2   PROOF OF PROPOSITION 4.2

We first introduce the following lemma before proving Proposition 4.2.

**Lemma B.1.** *Let $\mathcal{S}$ be a measure space and $P$ be a probability measure on $\mathcal{S}$. We further let $f : \mathcal{S} \to \mathbb{R}$ be any measure function and $c : \mathcal{S} \times \mathcal{S} \to \mathbb{R}_{\ge 0}$ be a cost function. Then for any scalar $\lambda \ge 0$, the following equality holds*

$$\inf_{\hat{P} \sim \mathcal{P}(\mathcal{S})} \left( \mathbb{E}_{\hat{s} \sim \hat{P}}[f(\hat{s})] + \lambda \mathcal{W}(\hat{P}, P) \right) = \mathbb{E}_{s \sim P} \left[ \inf_{\hat{s} \sim \mathcal{S}} (f(\hat{s}) + \lambda c(s, \hat{s})) \right], \tag{12}$$

*where $\mathcal{P}(\mathcal{S})$ represents all probability measures on $\mathcal{S}$, and $\mathcal{W}$ is the Wasserstein distance w.r.t. the cost function $c$.*

*Proof.* We prove this lemma by showing the left-hand-side (LHS) of Equation 12 is equivalent to its right-hand-side (RHS).

According to the definition of Wasserstein distance, the LHS could be written as

$$\text{LHS} = \inf_{\hat{P} \in \mathcal{P}(\mathcal{S})} \left( \mathbb{E}_{\hat{s} \sim \hat{P}}[f(\hat{s})] + \lambda \inf_{\gamma \in \Gamma(P, \hat{P})} \mathbb{E}_{(s, \hat{s}) \sim \gamma}[c(s, \hat{s})] \right).$$

The optimization of $\hat{P}$ and the inner optimization over $\gamma \in \Gamma(P, \hat{P})$ could be combined into a single optimization over all couplings $\gamma$ whose first marginal is $P$, and the second marginal, $\hat{P}$, could be arbitrary in $\mathcal{P}(\mathcal{S})$. We then have

$$\begin{aligned} \text{LHS} &= \inf_{\gamma \in \Gamma(P, \hat{P})} \left( \mathbb{E}_{\hat{s} \sim \hat{P}}[f(\hat{s})] + \lambda \mathbb{E}_{(s, \hat{s}) \sim \gamma}[c(s, \hat{s})] \right) \\ &= \inf_{\gamma \in \Gamma(P, \hat{P})} \mathbb{E}_{(s, \hat{s}) \sim \gamma} \left[ f(\hat{s}) + \lambda c(s, \hat{s}) \right], \end{aligned} \tag{13}$$

where the second equality holds by the linearity of expectation.

By the disintegration theorem for measures, any coupling $\gamma \in \Gamma(P, \hat{P})$ could be represented as the product of its first marginal $P$ and a stochastic kernel $K(d\hat{s}|s) : \mathcal{S} \to \mathcal{P}(\mathcal{S})$ such that

$$\gamma(ds, d\hat{s}) = K(d\hat{s}|s)P(ds). \tag{14}$$

This implies that optimizing over all couplings $\gamma \in \Gamma(P, \hat{P})$ is equivalent to optimizing over all possible stochastic kernels $K$. We substitute Equation 14 into Equation 13 to obtain

$$\begin{aligned} \text{LHS} &= \inf_{K} \int_{\mathcal{S}} \int_{\mathcal{S}} \left[ f(\hat{s}) + \lambda c(s, \hat{s}) \right] K(d\hat{s}|s)P(ds) \\ &= \inf_{K} \mathbb{E}_{s \sim P} \left[ \mathbb{E}_{\hat{s} \sim K(\cdot|s)} \left[ f(\hat{s}) + \lambda c(s, \hat{s}) \right] \right]. \end{aligned}$$

We change the position of the infimum operator and the inner expectation to get

$$\text{LHS} = \mathbb{E}_{s \sim P} \left[ \inf_{K(\cdot|s) \in \mathcal{P}(\mathcal{S})} \mathbb{E}_{\hat{s} \sim K(\cdot|s)} \left[ f(\hat{s}) + \gamma c(s, \hat{s}) \right] \right]. \tag{15}$$

We then solve the inner minimization problem for a fixed $s \in \mathcal{S}$:

$$\inf_{K(\cdot|s) \in \mathcal{P}(\mathcal{S})} \mathbb{E}_{\hat{s} \sim K(\cdot|s)} \left[ f(\hat{s}) + \gamma c(s, \hat{s}) \right].$$

Let $g_s(\hat{s}) \triangleq f(\hat{s}) + \gamma c(s, \hat{s})$. The problem is to find a probability measure $K(\cdot|s)$ that minimizes the expectation of $g_s(\hat{s})$. It is obvious that this minimum is achieved by concentrating the entire probability mass on point $\hat{s}$ where $g_s(\hat{s})$ attains its infimum.

Let $\hat{s}^{\star} = \arg\inf_{\hat{s} \in \mathcal{S}} g_s(\hat{s})$. The optimal measure is a Dirac measure $\delta_{\hat{s}^{\star}}$ centered on $\hat{s}^{\star}$. Therefore, we have

$$\begin{aligned} &\inf_{K(\cdot|s) \in \mathcal{P}(\mathcal{S})} \mathbb{E}_{\hat{s} \sim K(\cdot|s)} \left[ f(\hat{s}) + \gamma c(s, \hat{s}) \right] \\ &= \mathbb{E}_{\hat{s} \sim \delta_{\hat{s}^{\star}}} [g_s(\hat{s})] \\ &= f(\hat{s}^{\star}) + \gamma c(s, \hat{s}^{\star}) \\ &= \inf_{\hat{s} \in \mathcal{S}} \left[ f(\hat{s}) + \gamma c(s, \hat{s}) \right]. \end{aligned} \tag{16}$$

Finally, substituting Equation 16 back into Equation 15, we obtain the RHS of the lemma as

$$\begin{aligned} \text{LHS} &= \mathbb{E}_{s \sim P} \left[ \inf_{\hat{s} \in \mathcal{S}} \left( f(\hat{s}) + \gamma c(s, \hat{s}) \right) \right] \\ &= \text{RHS}. \end{aligned}$$

This concludes the proof. $\qquad\square$

Now we give our formal proof for Proposition 4.2. We restate it as follows.

**Proposition B.1** (Proposition 4.2). *Letting $\mathcal{M}_\epsilon$ be the Wasserstein uncertainty set defined by Equation 1, then the term $\inf_{\widehat{\mathcal{M}} \in \mathcal{M}_\epsilon} \mathbb{E}_{s' \sim P_{\widehat{\mathcal{M}}}} \left[ \max_{a' \sim \hat{\mu}(\cdot|s')} Q(s', a') \right]$ in Equation 2 is equivalent to*

$$\mathbb{E}_{s' \sim P_\mathcal{M}} \left[ \inf_{\bar{s}} \max_{a' \sim \hat{\mu}(\cdot|\bar{s})} Q(\bar{s}, a') \right], \quad s.t. \quad d(s', \bar{s}) \leq \epsilon. \tag{17}$$

*Proof.* The original term (OT) is a constrained optimization problem that can be solved by the Lagrange multiplier method. Let $\widehat{V}(s) = \max_{a \sim \hat{\mu}(\cdot|s)} Q(s, a)$. Then we define the Lagrange function as

$$\mathcal{L}(\widehat{\mathcal{M}}, \lambda) = \mathbb{E}_{\hat{s}' \sim \widehat{\mathcal{M}}(\cdot|s,a)} \widehat{V}(\hat{s}') + \lambda \left( \mathcal{W}(\widehat{\mathcal{M}}, \mathcal{M}) - \epsilon \right).$$

The LHS is equivalent to solving the dual problem:

$$\text{OT} = \sup_{\lambda \geq 0} \inf_{\widehat{\mathcal{M}}} \mathcal{L}(\widehat{\mathcal{M}}, \lambda). \tag{18}$$

For a fixed $\lambda$, we solve the inner minimization problem $\inf_{\widehat{M}} \mathcal{L}(\widehat{M}, \lambda)$:

$$\inf_{\widehat{\mathcal{M}}} \mathcal{L}(\widehat{\mathcal{M}}, \lambda) = \inf_{\widehat{\mathcal{M}}} \left( \mathbb{E}_{\hat{s}' \sim \widehat{\mathcal{M}}(\cdot|s,a)} \widehat{V}(\hat{s}') + \lambda \mathcal{W}(\widehat{\mathcal{M}}, \mathcal{M}) \right) - \lambda \epsilon. \tag{19}$$

According to Lemma B.1, we have

$$\inf_{\widehat{\mathcal{M}}} \left( \mathbb{E}_{\hat{s}' \sim \widehat{\mathcal{M}}(\cdot|s,a)} \widehat{V}(\hat{s}') + \lambda \mathcal{W}(\widehat{\mathcal{M}}, \mathcal{M}) \right) = \mathbb{E}_{s' \sim \mathcal{M}(\cdot|s,a)} \left[ \inf_{\hat{s}'} \left( \widehat{V}(\hat{s}') + \lambda c(s', \hat{s}') \right) \right]. \tag{20}$$

Substituting Equation 20 into Equation 19 and Equation 18, we obtain the dual reformulation of the original term as

$$\text{OT} = \sup_{\lambda \geq 0} \left\{ \mathbb{E}_{s' \sim T(\cdot|s,a)} \left[ \inf_{\hat{s}'} \left( \hat{V}(\hat{s}') + \lambda c(s', \hat{s}') \right) \right] - \lambda \epsilon \right\}. \tag{21}$$

The RHS in Equation 21 is exactly the Lagrange dual reformulation of Equation 17. This implies Equation 17 holds, which concludes the proof. $\qquad \square$

## B.3 PROOF OF PROPOSITION 4.3

*Proof.* We only discuss the case where $s' \sim P_{\text{src}}(\cdot|s,a)$, since for $s' \sim P_{\text{tar}}(\cdot|s,a)$, the proof is identical as Proposition 4.1. For $s' \sim P_{\text{src}}(\cdot|s,a)$, letting $Q_1$ and $Q_2$ be two arbitrary $Q$ functions, we have

$$\|\mathcal{T}_{\text{RCB}} Q_1 - \mathcal{T}_{\text{RCB}} Q_2\|_\infty = \gamma \max_{s,a} \left| \mathbb{E}_{s'} \left[ \inf_{\bar{s} \in U_\epsilon(s')} \max_{a' \sim \hat{\mu}(\cdot|\bar{s})} Q_1(\bar{s}, a') - \inf_{\bar{s} \in U_\epsilon(s')} \max_{a' \sim \hat{\mu}(\cdot|\bar{s})} Q_2(\bar{s}, a') \right] \right|$$

$$\leq \gamma \max_{s,a} \mathbb{E}_{s'} \left| \inf_{\bar{s} \in U_\epsilon(s')} \max_{a' \sim \hat{\mu}(\cdot|\bar{s})} Q_1(\bar{s}, a') - \inf_{\bar{s} \in U_\epsilon(s')} \max_{a' \sim \hat{\mu}(\cdot|\bar{s})} Q_2(\bar{s}, a') \right|$$

$$\leq \gamma \max_{s,a} \|Q_1 - Q_2\|_\infty$$

$$= \gamma \|Q_1 - Q_2\|_\infty,$$

where the second inequality holds from Equation 10 and Equation 11. Then, we can conclude that the practical RCB operator is still a $\gamma$-contraction operator. $\qquad \square$

## B.4 PROOF OF PROPOSITION 4.4

*Proof.* For any $(s, a) \in \mathcal{D}_{\text{src}}$,

$$\mathcal{T}_{\text{RCB}} Q(s, a) - \mathcal{T} Q(s, a)$$

$$= \gamma \left( \mathbb{E}_{s' \sim P_{\mathcal{M}_{\text{src}}}} \left[ \inf_{\bar{s} \in U_\epsilon(s')} \max_{a' \sim \hat{\mu}(\cdot|\bar{s})} Q(\bar{s}, a') \right] - \mathbb{E}_{s' \sim P_{\mathcal{M}_{\text{tar}}}} \left[ \max_{a' \sim \hat{\mu}(\cdot|s')} Q(s', a') \right] \right)$$

$$\leq \gamma \left( \inf_{s' \sim P_{\mathcal{M}_{\text{tar}}}} \max_{a' \sim \hat{\mu}(\cdot|s')} Q(s', a') - \mathbb{E}_{s' \sim P_{\mathcal{M}_{\text{tar}}}} \left[ \max_{a' \sim \hat{\mu}(\cdot|s')} Q(s', a') \right] \right)$$

$$\leq 0,$$

where the first inequality holds by $\text{supp}(P_{\text{tar}}(\cdot|s, a)) \subseteq U_\epsilon(s'_{\text{src}})$. In the mean time, for any $(s, a) \in \mathcal{D}_{\text{src}}$, we have

$$\mathcal{T}_{\text{RCB}} Q(s, a) - \mathcal{T} Q(s, a)$$

$$= \gamma \left( \mathbb{E}_{s' \sim P_{\mathcal{M}_{\text{src}}}} \left[ \inf_{\bar{s} \in U_\epsilon(s')} \max_{a' \sim \hat{\mu}(\cdot|\bar{s})} Q(\bar{s}, a') \right] - \mathbb{E}_{s' \sim P_{\mathcal{M}_{\text{tar}}}} \left[ \max_{a' \sim \hat{\mu}(\cdot|s')} Q(s', a') \right] \right)$$

$$= \gamma \left( \mathbb{E}_{\bar{s}} \left[ \max_{a' \sim \hat{\mu}(\cdot|\bar{s})} Q(\bar{s}, a') \right] - \mathbb{E}_{s' \sim P_{\mathcal{M}_{\text{tar}}}} \left[ \max_{a' \sim \hat{\mu}(\cdot|s')} Q(s', a') \right] \right)$$

$$\geq \gamma \left( \min_{\bar{s}} \left[ \max_{a' \sim \hat{\mu}(\cdot|\bar{s})} Q(\bar{s}, a') \right] - \max_{s' \sim P_{\mathcal{M}_{\text{tar}}}} \left[ \max_{a' \sim \hat{\mu}(\cdot|s')} Q(s', a') \right] \right).$$

Letting $\bar{s}^\star = \arg\min_{\bar{s}} \left[ \max_{a' \sim \hat{\mu}(\cdot|\bar{s})} Q(\bar{s}, a') \right]$ and $s^\star = \arg\max_{s' \sim P_{\mathcal{M}_{\text{tar}}}} \left[ \max_{a' \sim \hat{\mu}(\cdot|s')} Q(s', a') \right]$, we have

$$\mathcal{T}_{\text{RCB}} Q(s, a) - \mathcal{T} Q(s, a)$$

$$\geq \gamma \left( \max_{a' \sim \hat{\mu}(\cdot|\bar{s}^\star)} Q(\bar{s}, a') - \max_{a' \sim \hat{\mu}(\cdot|s^\star)} Q(s^\star, a') \right)$$

$$\geq \gamma \left( Q(\bar{s}, a^\star) - Q(s^\star, a^\star) \right)$$

$$\geq -2\gamma\epsilon K_Q,$$

where $a^\star = \arg\max_{a' \sim \hat{\mu}(\cdot|s^\star)} Q(s^\star, a')$, and the last inequality holds by the Lipschitz continuity assumption and triangle inequality.

Combining the above results, we have

$$\mathcal{T} Q(s, a) - 2\gamma\epsilon K_Q \leq \mathcal{T}_{\text{RCB}} Q(s, a) \leq \mathcal{T} Q(s, a). \tag{22}$$

Let $Q^k$ denote the $Q$ value at iteration $k$, and let the initial $Q$ value be $Q^0$. After one iteration using the RCB operator and the oracle optimal Bellman operator, according to Equation 22,

$$Q^1(s, a) - \frac{2\gamma\epsilon K_Q}{1 - \gamma} (1 - \gamma) \leq \hat{Q}^1_{\text{RCB}}(s, a) \leq Q^1(s, a).$$

Suppose when $k = i$, we have

$$Q^i(s, a) - \frac{2\gamma\epsilon K_Q}{1 - \gamma} (1 - \gamma^i) \leq \hat{Q}^i_{\text{RCB}}(s, a) \leq Q^i(s, a), \quad i \in \mathbb{Z}^+. \tag{23}$$

For $k = i + 1$, we have

$$\mathcal{T} \hat{Q}^i_{\text{RCB}}(s, a) - 2\gamma\epsilon K_Q \leq \hat{Q}^{i+1}_{\text{RCB}}(s, a) = \mathcal{T}_{\text{RCB}} \hat{Q}^i_{\text{RBC}}(s, a) \leq \mathcal{T} \hat{Q}^i_{\text{RCB}}(s, a).$$

On the one hand, we have

$$\mathcal{T} \hat{Q}^i_{\text{RCB}}(s, a)$$

$$\geq \mathcal{T} \left( Q^i(s, a) - \frac{2\gamma\epsilon K_Q}{1 - \gamma} (1 - \gamma^i) \right)$$

$$= r(s, a) + \gamma \mathbb{E}_{s' \sim P_{\text{tar}}} \left[ \max_{a' \sim \hat{\mu}(\cdot|s')} \left( Q^i(s', a') - \frac{2\gamma\epsilon K_Q}{1 - \gamma} (1 - \gamma^i) \right) \right]$$

$$= r(s, a) + \gamma \mathbb{E}_{s' \sim P_{\text{tar}}} \left[ \max_{a' \sim \hat{\mu}(\cdot|s')} Q^i(s', a') \right] - \gamma \frac{2\gamma\epsilon K_Q}{1 - \gamma} (1 - \gamma^i)$$

$$= \mathcal{T} Q^i(s, a) - \gamma \frac{2\gamma\epsilon K_Q}{1 - \gamma} (1 - \gamma^i)$$

$$= Q^{i+1}(s, a) - \gamma \frac{2\gamma\epsilon K_Q}{1 - \gamma} (1 - \gamma^i).$$

Therefore, we have

$$\hat{Q}^{i+1}_{\text{RCB}}(s, a)$$

$$\geq Q^{i+1}(s, a) - \gamma \frac{2\gamma\epsilon K_Q}{1 - \gamma} (1 - \gamma^i) - 2\gamma\epsilon K_Q \tag{24}$$

$$= Q^{i+1}(s, a) - \frac{2\gamma\epsilon K_Q}{1 - \gamma} (1 - \gamma^{i+1}).$$

On the other hand,

$$\mathcal{T}\hat{Q}_{\text{RCB}}^{i}(s,a) \leq \mathcal{T}Q^{i}(s,a) = Q^{i+1}(s,a).$$

Therefore, we have

$$\hat{Q}_{\text{RCB}}^{i+1}(s,a) \leq Q^{i+1}(s,a). \tag{25}$$

Combining the results of Equation 24 and Equation 25, we have

$$Q^{i+1}(s,a) - \frac{2\gamma\epsilon K_Q}{1-\gamma}(1-\gamma^{i+1}) \leq \hat{Q}_{\text{RCB}}^{i+1}(s,a) \leq Q^{i+1}(s,a).$$

Hence, Equation 23 still holds for $k = i + 1$. Therefore, Equation 23 holds for all $k \in \mathbb{Z}^+$. If $k$ is large enough, such that $\hat{Q}_{\text{RCB}}$ and $Q(s,a)$ converge to the fixed point, then we have

$$Q_{\hat{\mu}}^{\star}(s,a) - \frac{2\gamma\epsilon K_Q}{1-\gamma} \leq \hat{Q}_{\text{RCB}}(s,a) \leq Q_{\hat{\mu}}^{\star}(s,a),$$

which concludes the proof. $\qquad\square$

### B.5 PROOF OF PROPOSITION 4.5

*Proof.* The learned value function $\widehat{V}_{\text{RCB}}(s)$ by repeatedly applying $\mathcal{T}_{\text{RCB}}$ satisfies:

$$\widehat{V}_{\text{RCB}}(s) = r(s,a^{\star}) + \gamma\mathbb{E}_{s'\sim P_{\text{src}}}\left[\inf_{\bar{s}\in U_{\epsilon}(s')}\widehat{V}_{\text{RCB}}(\bar{s})\right]$$

where $a^{\star} = \pi_{\text{RCB}}(\cdot|s) = \arg\max_{a\sim\hat{\mu}(\cdot|s)}\left[r(s,a) + \gamma\mathbb{E}_{s'\sim P_{\text{src}}}\left[\inf_{\bar{s}\in U_{\epsilon}(s')}\widehat{V}_{\text{RCB}}(\bar{s})\right]\right]$.

Let $c = \max\left\{c\,|\,U_c(s'_{\text{tar}}) \subseteq U_{\epsilon}(s'_{\text{src}}), s'_{\text{tar}} \sim P_{\text{tar}}(\cdot|s,a), (s,a,s'_{\text{src}}) \sim \mathcal{D}_{\text{src}}\right\}$. Then we have

$$\mathbb{E}_{s'\sim P_{\text{tar}}}\left[\inf_{\bar{s}\in U_c(s')}\widehat{V}_{\text{RCB}}(\bar{s})\right] \geq \mathbb{E}_{s'\sim P_{\text{src}}}\left[\inf_{\bar{s}\in U_{\epsilon}(s')}\widehat{V}_{\text{RCB}}(\bar{s})\right],$$

since $U_{\epsilon}(s'_{\text{src}})$ has a broader region than $U_c(s'_{\text{tar}})$. Given any dynamics $P$ which satisfies $\mathcal{W}(P(\cdot|s,a), P_{\text{tar}}(\cdot|s,a)) \leq c$, we can iteratively evaluate $\pi_{\text{RCB}}$ within $P$:

$$\begin{aligned} V^{k+1}(s) &= \mathcal{T}_P^{\pi_{\text{RCB}}}(V^k(s)) \\ &= r(s,a\sim\pi_{\text{RCB}}(\cdot|s)) + \gamma\mathbb{E}_{s'\sim P(\cdot|s,a)}\left(V^k(s')\right) \\ &\geq r(s,a\sim\pi_{\text{RCB}}(\cdot|s)) + \gamma\mathbb{E}_{s'\sim P_{\text{tar}}}\left[\inf_{\bar{s}\in U_c(s')}V^k(\bar{s})\right]. \end{aligned}$$

If we initialize $V^0(s)$ as $\widehat{V}_{\text{RCB}}(s)$, we have

$$\begin{aligned} V^1(s) &= \mathcal{T}_P^{\pi_{\text{RCB}}}(V^0(s)) \\ &\geq r(s,a\sim\pi_{\text{RCB}}(\cdot|s)) + \gamma\mathbb{E}_{s'\sim P_{\text{tar}}}\left[\inf_{\bar{s}\in U_c(s')}\widehat{V}_{\text{RCB}}(\bar{s})\right] \\ &\geq r(s,a\sim\pi_{\text{RCB}}(\cdot|s)) + \gamma\mathbb{E}_{s'\sim P_{\text{src}}}\left[\inf_{\bar{s}\in U_{\epsilon}(s')}\widehat{V}_{\text{RCB}}(\bar{s})\right] \\ &= \widehat{V}_{\text{RCB}}(s) \\ &= V^0(s). \end{aligned}$$

According to the monotonicity of the Bellman operator, we have $V^2(s) = \mathcal{T}_P^{\pi_{\text{RCB}}}(V^1(s)) \geq \mathcal{T}_P^{\pi_{\text{RCB}}}(V^0(s)) = V^1(s)$. Similarly, we can get $V^k(s) \geq V^{k-1}(s) \geq \cdots \geq V^0(s) = \hat{V}_{\text{RCB}}(s)$. Given that $\mathcal{T}_P^{\pi_{\text{RCB}}}$ is a $\gamma$-contraction, $V_P^{\pi_{\text{RCB}}}(s) = \lim_{k\to\infty}V^k(s) \geq \widehat{V}_{\text{RCB}}(s)$, which proves Equation 3 and conclude the proof. $\qquad\square$

### B.6 Proof of Proposition 4.6

*Proof.* We draw the proof inspiration from (Lyu et al., 2022). Given that $\sup_{s,a} D_{TV}(\widehat{P}_{\text{tar}}(\cdot|s,a), P_{\text{tar}}(\cdot|s,a)) \le \epsilon < \frac{1}{2}$, we have

$$
\begin{aligned}
1 > 2\epsilon \\
&\ge 2 \sup_{s,a} D_{TV}(\widehat{P}_{\text{tar}}(\cdot|s,a), P_{\text{tar}}(\cdot|s,a)) \\
&\ge \sum_{s'} \left| \widehat{P}_{\text{tar}}(s'|s,a) - P_{\text{tar}}(s'|s,a) \right| \\
&= \sum_{s' \in \text{supp}(P_{\text{tar}}(\cdot|s,a))} \left| \widehat{P}_{\text{tar}}(s'|s,a) - P_{\text{tar}}(s'|s,a) \right| + \sum_{s' \notin \text{supp}(P_{\text{tar}}(\cdot|s,a))} \left| \widehat{P}_{\text{tar}}(s'|s,a) - P_{\text{tar}}(s'|s,a) \right| \\
&\ge \sum_{s' \notin \text{supp}(P_{\text{tar}}(\cdot|s,a))} \widehat{P}_{\text{tar}}(s'|s,a).
\end{aligned}
$$

Note that the maximum $Q$ value $Q_{\max} \le \frac{r_{\max}}{1-\gamma}$. Thus, we have

$$
\begin{aligned}
\inf_{\{s_i'\}^N \sim \widehat{P}_{\text{tar}}(\cdot|s,a)} & \left[ \max_{a_i' \sim \hat{\mu}(\cdot|s_i')} Q(s_i', a_i') \right] \\
&\le \mathbb{E}_{\{s_i'\}^N \sim \widehat{P}_{\text{tar}}(\cdot|s,a)} \left[ \max_{a_i' \sim \hat{\mu}(\cdot|s_i')} Q(s_i', a_i') \right] \\
&\le \mathbb{P}\left( \bigcap_i \{s' \in \text{supp}(P_{\text{tar}}(\cdot|s,a))\} \right) \cdot \mathbb{E}_{s' \sim P_{\text{tar}}(\cdot|s,a)} \left[ \max_{a' \sim \hat{\mu}(\cdot|s')} Q(s', a') \right] \\
&\quad + \mathbb{P}\left( \bigcup_i \{s' \notin \text{supp}(P_{\text{tar}}(\cdot|s,a))\} \right) \cdot Q_{\max} \\
&\le \mathbb{E}_{s' \sim P_{\text{tar}}(\cdot|s,a)} \left[ \max_{a' \sim \hat{\mu}(\cdot|s')} Q(s', a') \right] + \left( 1 - (\mathbb{P}(s_1' \in \text{supp}(P_{\text{tar}}(\cdot|s,a))))^N \right) \frac{r_{\max}}{1-\gamma} \\
&\le \mathbb{E}_{s' \sim P_{\text{tar}}(\cdot|s,a)} \left[ \max_{a' \sim \hat{\mu}(\cdot|s')} Q(s', a') \right] + \left( 1 - (1-2\epsilon)^N \right) \frac{r_{\max}}{1-\gamma},
\end{aligned}
$$

where the first inequality uses the law of total expectation. Thus, we conclude the proof. $\square$

## C  Experimental Settings

In this section, we introduce the detailed environmental settings missing in the main text.

### C.1  Tasks and Datasets

**Target domain and datasets.** We directly adopt the four locomotion tasks from MuJoCo Engine (Todorov et al., 2012) as the target domain tasks: `halfcheetah-v2`, `hopper-v2`, `walker2d-v2`, `ant-v3`. For the target domain datasets, we reuse the datasets in D4RL (Fu et al., 2020) for each task. Since cross-domain offline RL only allows a small quantity of target domain data, we sample $10\%$ data from the original D4RL datasets as the target domain datasets. The target domain datasets consist of four data qualities for each task: the **medium** datasets that contain samples collected by an early-stopped SAC policy; the **medium-replay** datasets that represent the replay buffer of the medium-level SAC agent; the **medium-expert** datasets that mix the medium data and expert data at a 50-50 ratio; the **expert** datasets that are collected by an SAC policy trained to the expert level. The trained policy is evaluated in the target domain, and the evaluation metric we use is `Normalized Score` in D4RL:

$$
\text{Normalized Score} = \frac{J_\pi - J_{\text{random}}}{J_{\text{expert}} - J_{\text{random}}} \times 100\%,
$$

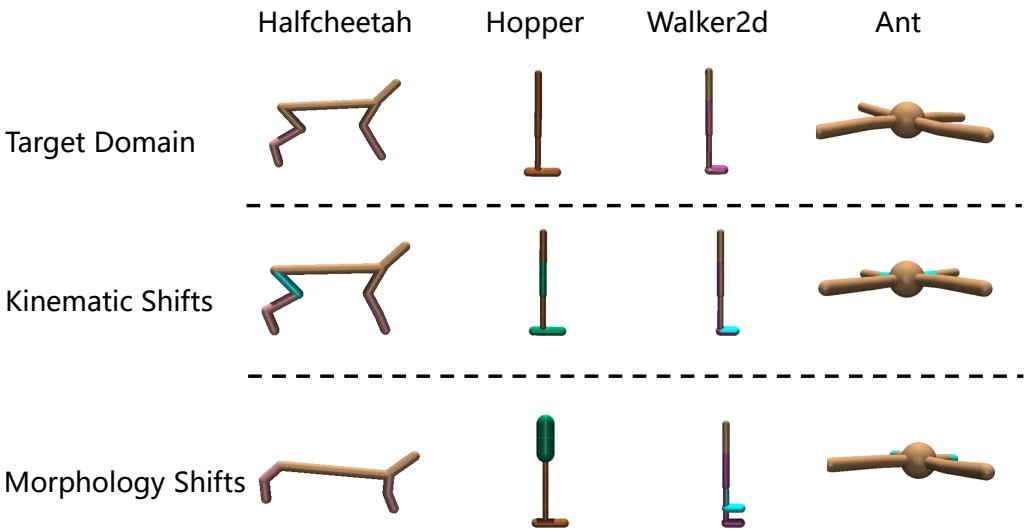

Figure 4: Visualization of the target domains and source domains with kinematic shifts and morphology shifts, across four tasks (`halfcheetah`, `hopper`, `walker2d`, `ant`).

where $J_\pi$ is the return acquired by the trained policy in the target domain, and $J_{\text{expert}}$ and $J_{\text{random}}$ are the returns acquired by the expert policy and the random policy in the target domain, respectively.

**Source domain and datasets.** To simulate the source domain with dynamics shifts, we consider the four MuJoCo tasks (`halfcheetah-v2`, `hopper-v2`, `walker2d-v2`, `ant-v3`) with kinematic shifts and morphology shifts introduced as the source domain. The kinematic shifts refer to some joints of the robot being broken and unable to rotate, while the morphology shifts indicate that the robot's morphology is modified, differing from the target domain. To illustrate this more clearly, we visualize the robots in both the target and source domains for all four tasks in Figure 4. We also provide detailed code-level modifications for implementing the dynamics shifts in the following section.

For the source domain datasets, we follow a data collection process similar to D4RL. Specifically, we train an SAC policy in the source domain for 1M environmental steps and log policy checkpoints at different steps for trajectory rollouts. The **medium** datasets are collected using a logged policy that achieves approximately half the performance of the expert policy. The **medium-replay** datasets consist of the logged replay buffer from the medium-level agent. The **expert** datasets are collected using the final policy checkpoint, while the **medium-expert** datasets are a 50-50 mixture of medium-level and expert-level data. Note that all the source domain datasets contain about 1M samples, whereas the target domain datasets contain much fewer samples.

## C.2 KINEMATIC SHIFTS REALIZATION

To simulate the kinematic shifts in the source domain, we modify the `xml` files of the original environments. Specifically, we change the rotation angle of some joints of the simulated robot for different tasks:

*halfcheetah-kinematic*: The rotation angle of the joint on the thigh of the robot's back leg is modified from $[-0.52, 1.05]$ to $[-0.0052, 0.0105]$.

```
# broken back thigh joint
<joint axis="0 1 0" damping="6" name="bthigh" pos="0 0 0" range="
    -.0052 .0105" stiffness="240" type="hinge"/>
```

*hopper-kinematic*: The rotation angle of the head joint is modified from $[-150, 0]$ to $[-0.15, 0]$ and the rotation angle of the foot joint is modified from $[-45, 45]$ to $[-18, 18]$.

```
# broken head joint
<joint axis="0 -1 0" name="thigh_joint" pos="0 0 1.05" range="
    -0.15 0" type="hinge"/>
# broken foot joint
<joint axis="0 -1 0" name="foot_joint" pos="0 0 0.1" range="-18 18
    " type="hinge"/>
```

*walker2d-kinematic*: The rotation angle of the right foot joint is modified from $[-45, 45]$ to $[-0.45, 0.45]$.

```
# broken right foot joint
<joint axis="0 -1 0" name="foot_joint" pos="0 0 0.1" range="-0.45
    0.45" type="hinge"/>
```

*ant-kinematic*: The rotation angles of the joints on the hip of two front legs are modified from $[-30, 30]$ to $[-0.3, 0.3]$.

```
# broken hip joints of front legs
<joint axis="0 0 1" name="hip_1" pos="0.0 0.0 0.0" range="-0.3 0.3
    " type="hinge"/>
<joint axis="0 0 1" name="hip_2" pos="0.0 0.0 0.0" range="-0.3 0.3
    " type="hinge"/>
```

## C.3 MORPHOLOGY SHIFTS REALIZATION

Akin to the kinematic shifts, we modify the `xml` files to simulate morphology shifts:

*halfcheetah-morph*: The sizes of the back thigh and the forward thigh are modified.

```
# back thigh
<geom fromto="0 0 0 -0.0001 0 -0.0001" name="bthigh" size="0.046"
    type="capsule"/>
<body name="bshin" pos="-0.0001 0 -0.0001">
# front thigh
<geom fromto="0 0 0 0.0001 0 0.0001" name="fthigh" size="0.046"
    type="capsule"/>
<body name="fshin" pos="0.0001 0 0.0001">
```

*hopper-morph*: The head size of the robot is modified.

```
# head size
<geom friction="0.9" fromto="0 0 1.45 0 0 1.05" name="torso_geom"
    size="0.125" type="capsule"/>
```

*walker2d-morph*: The thigh on the right leg of the robot is modified.

```
# right leg
<body name="thigh" pos="0 0 1.05">
<joint axis="0 -1 0" name="thigh_joint" pos="0 0 1.05" range="-150
    0" type="hinge"/>
<geom friction="0.9" fromto="0 0 1.05 0 0 1.045" name="thigh_geom"
    size="0.05" type="capsule"/>
<body name="leg" pos="0 0 0.35">
  <joint axis="0 -1 0" name="leg_joint" pos="0 0 1.045" range="
    -150 0" type="hinge"/>
  <geom friction="0.9" fromto="0 0 1.045 0 0 0.3" name="leg_geom"
    size="0.04" type="capsule"/>
```

```
  <body name="foot" pos="0.2 0 0">
    <joint axis="0 -1 0" name="foot_joint" pos="0 0 0.3" range="-45
        45" type="hinge"/>
    <geom friction="0.9" fromto="-0.0 0 0.3 0.2 0 0.3" name="
        foot_geom" size="0.06" type="capsule"/>
  </body>
</body>
</body>
```

***ant-morph***: The size of the robot's two front legs is reduced.

```
# front leg 1
<geom fromto="0.0 0.0 0.0 0.1 0.1 0.0" name="left_ankle_geom" size
    ="0.08" type="capsule"/>
# front leg 2
<geom fromto="0.0 0.0 0.0 -0.1 0.1 0.0" name="right_ankle_geom"
    size="0.08" type="capsule"/>
```

## D  IMPLEMENTATION DETAILS

In this section, we provide the implementation details for the baselines we use in our experiments and our method, DROCO.

### D.1  BASELINES

**IQL$^\star$:** IQL$^\star$ is the cross-domain adaptation of IQL (Kostrikov et al., 2021). IQL$^\star$ follows the same algorithmic procedure except being trained on both target and source domain datasets. The state value function is trained by expectile regression:

$$\mathcal{L}_V = \mathbb{E}_{(s,a)\sim\mathcal{D}_{\text{src}}\cup\mathcal{D}_{\text{tar}}}\left[L_2^\tau(Q_{\theta'}(s,a) - V_\psi(s))\right],$$

where $L_2^\tau(u) = |\tau - \mathbb{I}(u < 0)|\, u^2$, $\mathbb{I}(\cdot)$ is the indicator function, and $\theta'$ is the parameter of the target network. This expectile regression enables learning an in-sample optimal value function. Subsequently, the state-action value function is updated by:

$$\mathcal{L}_Q = \mathbb{E}_{(s,a,r,s')\sim\mathcal{D}_{\text{src}}\cup\mathcal{D}_{\text{tar}}}\left[(r(s,a) + \gamma V_\psi(s') - Q_\theta(s,a))^2\right].$$

Then the advantage value is computed as $A(s,a) = Q(s,a) - V(s,a)$. Based on this, the policy is obtained through exponential advantage-weighted behavior cloning:

$$\mathcal{L}_\pi = -\mathbb{E}_{(s,a)\sim\mathcal{D}_{\text{src}}\cup\mathcal{D}_{\text{tar}}}\left[\exp(\beta \times A(s,a))\log\pi_\phi(a|s)\right],$$

where $\beta$ is the inverse temperature coefficient. We implement IQL$^\star$ based on the official codebase[2] of IQL.

**CQL$^\star$:** the cross-domain version of CQL (Kumar et al., 2020) similar to IQL$^\star$. CQL learns a conservative value function that lower bounds the true value function:

$$\mathcal{L}_Q = \alpha\mathbb{E}_{s\in\mathcal{D}}\left[\log\sum_a \exp(Q(s,a)) - \mathbb{E}_{a\sim\mu}[Q(s,a)]\right] + \frac{1}{2}\mathbb{E}_{(s,a,s')\in\mathcal{D}}\left[(Q(s,a) - \mathcal{T}Q(s,a))^2\right]$$

The policy $\pi$ is then optimized with SAC (Haarnoja et al., 2018). We implement CQL$^\star$ based on the implementation of CORL[3].

**BOSA:** BOSA (Liu et al., 2024a) identifies two key challenges in cross-domain offline RL: the state-action OOD problem and the dynamics OOD problem. To address these, BOSA proposes two support constraints. Specifically, BOSA handles the OOD state-action problem by supported policy

---

[2]https://github.com/ikostrikov/implicit_q_learning.git
[3]https://github.com/tinkoff-ai/CORL.git

optimization, and mitigates the OOD dynamics problem by supported value optimization. The critic is updated through supported value optimization:

$$\mathcal{L}_Q = \mathbb{E}_{(s,a)\sim\mathcal{D}_{\text{src}}}\left[Q_{\theta_i}(s,a)\right] + \mathbb{E}_{\substack{(s,a,r,s')\sim\mathcal{D}_{\text{src}}\cup\mathcal{D}_{\text{tar}},\\ a'\sim\pi_\phi(s')}}\left[\mathbb{I}(\hat{P}_{tar}(s'|s,a) > \epsilon)(Q_{\theta_i}(s,a) - y)^2\right],$$

where $\mathbb{I}(\cdot)$ is the indicator function, and $\hat{P}_{\text{tar}}(s'|s,a)$ is the estimated target domain dynamics, and $\epsilon$ is the threshold coefficient. The policy in BOSA is updated by supported policy optimization to mitigate the OOD action issue:

$$\mathcal{L}_\pi = \mathbb{E}_{s\sim\mathcal{D}_{\text{src}}\cup\mathcal{D}_{\text{tar}},\ a\sim\pi_\phi(s)}\left[Q_{\theta_i}(s,a)\right], \quad \text{s.t. } \mathbb{E}_{s\sim\mathcal{D}_{\text{src}}\cup\mathcal{D}_{\text{tar}}}\left[\hat{\pi}_{\text{mix}}(\pi_\phi(s)\mid s)\right] > \epsilon',$$

where $\epsilon'$ is the threshold coefficient, and $\hat{\pi}_{\phi_{\text{mix}}}(\cdot|s)$ is the behavior policy of the mixed datasets $\mathcal{D}_{\text{src}}\cup\mathcal{D}_{\text{tar}}$ learned with CVAE (Kingma et al., 2013). We adopt the BOSA implementation from ODRL[4] benchmark (Lyu et al., 2024b), which provides reliable implementations for various off-dynamics RL algorithms.

**DARA.** DARA (Liu et al., 2022) employs dynamics-aware reward modification to achieve dynamics adaptation, extending DARC (Eysenbach et al., 2020) to the offline setting. Specifically, DARA trains two domain classifiers $q_{\theta_{SAS}}(\text{target}|s_t, a_t, s_{t+1})$ and $q_{\theta_{SA}}(\text{target}|s_t, a_t)$ as follows:

$$\mathcal{L}_{\theta_{SAS}} = \mathbb{E}_{\mathcal{D}_{\text{tar}}}\left[\log q_{\theta_{SAS}}(\text{target}|s_t, a_t, s_{t+1})\right] + \mathbb{E}_{\mathcal{D}_{\text{src}}}\left[\log(1 - q_{\theta_{SAS}}(\text{target}|s_t, a_t, s_{t+1}))\right]$$
$$\mathcal{L}_{\theta_{SA}} = \mathbb{E}_{\mathcal{D}_{\text{tar}}}\left[\log q_{\theta_{SA}}(\text{target}|s_t, a_t)\right] + \mathbb{E}_{\mathcal{D}_{\text{src}}}\left[\log(1 - q_{\theta_{SA}}(\text{target}|s_t, a_t))\right].$$

The domain classifiers are used to quantify the dynamics gap $\log\frac{P_{\mathcal{M}_{\text{tar}}}(s_{t+1}|s_t,a_t)}{P_{\mathcal{M}_{\text{src}}}(s_{t+1}|s_t,a_t)}$ between the source domain and the target domain according to Bayes' rule. Then the estimated dynamics gap serves as a penalty to the source domain rewards:

$$\hat{r}_{\text{DARA}} = r - \lambda \times \delta_r, \quad \delta_r(s_t, a_t) = -\log\frac{q_{\theta_{\text{SAS}}}(\text{target}|s_t, a_t, s_{t+1})q_{\theta_{\text{SA}}}(\text{source}|s_t, a_t)}{q_{\theta_{\text{SAS}}}(\text{source}|s_t, a_t, s_{t+1})q_{\theta_{\text{SA}}}(\text{target}|s_t, a_t)}, \quad (26)$$

where $\lambda$ controls the intensity of the reward penalty. We use the DARA implementation from ODRL and follow the hyperparameter setting in the original paper: $\lambda$ is set to $0.1$, and the reward penalty is clipped within $[-10, 10]$ for training stability.

**IGDF.** IGDF (Wen et al., 2024) quantifies the domain discrepancy between the source domain and the target domain with contrastive representation learning. To facilitate effective knowledge transfer, IGDF implements data filtering to selectively share source domain samples exhibiting smaller dynamics gaps. Specifically, IGDF trains a score function $h(\cdot)$ using $(s, a, s'_{\text{tar}}) \sim \mathcal{D}_{\text{tar}}$ as the positive samples, and transitions $(s, a, s'_{\text{src}})$ as the negative samples, where $(s, a) \sim \mathcal{D}_{\text{tar}}$ and $s'_{\text{src}} \sim \mathcal{D}_{\text{src}}$. $h(\cdot)$ is optimized via the following contrastive learning objective:

$$\mathcal{L} = -\mathbb{E}_{(s,a,s'_{\text{tar}})}\mathbb{E}_{s'_{\text{src}}}\left[\log\frac{h(s, a, s'_{\text{tar}})}{\sum_{s'\in s'_{\text{tar}}\cup s'_{\text{src}}} h(s, a, s')}\right].$$

Based on the learned score function, IGDF proposes to selectively share source domain data for training value functions:

$$\mathcal{L}_Q = \frac{1}{2}\mathbb{E}_{\mathcal{D}_{\text{tar}}}\left[(Q_\theta - \mathcal{T}Q_\theta)^2\right] + \frac{1}{2}\alpha \cdot h(s, a, s')\mathbb{E}_{(s,a,s')\sim\mathcal{D}_{\text{src}}}\left[\mathbb{I}(h(s, a, s') > h_{\xi\%})(Q_\theta - \mathcal{T}Q_\theta)^2\right],$$

where $\mathbb{I}(\cdot)$ is the indicator function, $\alpha$ is the weighting coefficient, $\xi$ is the data selection ratio. We implement IGDF based on its official codebase[5].

**OTDF.** OTDF (Lyu et al., 2025) estimates the discrepancy between the source domain and target domain by computing the Wasserstein distance (Peyré et al., 2019):

$$\mathcal{W}(u, u') = \min_{\mu\in M}\sum_{t=1}^{|\mathcal{D}_{\text{src}}|}\sum_{t'=1}^{|\mathcal{D}_{\text{tar}}|} C(u_t, u'_{t'})\cdot\mu_{t,t'}, \quad (27)$$

---

[4] https://github.com/OffDynamicsRL/off-dynamics-rl.git
[5] https://github.com/BattleWen/IGDF.git

where $u = s_{\text{src}} \oplus a_{\text{src}} \oplus s'_{\text{src}}$, $u' = s_{\text{tar}} \oplus a_{\text{tar}} \oplus s'_{\text{tar}}$ are the concatenated vectors, $C$ is the cost function and $M$ is the coupling matrices. After solving Equation 27 for the optimal coupling matrix $\mu^\star$, the OTDF computes the distance between a source domain sample and the target domain dataset via

$$d(u_t) = -\sum_{t'=1}^{|\mathcal{D}_{\text{tar}}|} C(u_t, u_{t'})\mu^\star_{t,t'}, \quad u_t = (s^t_{\text{src}}, a^t_{\text{src}}, (s'_{\text{src}})^t) \sim \mathcal{D}_{\text{src}}.$$

Then the critic is updated by

$$\mathcal{L}_Q = \mathbb{E}_{\mathcal{D}_{\text{tar}}}\left[(Q_\theta - \mathcal{T}Q_\theta)^2\right] + \mathbb{E}_{(s,a,s')\sim\mathcal{D}_{\text{src}}}\left[\exp(\alpha \times d)\mathbb{I}(d > d_\%)(Q_\theta - \mathcal{T}Q_\theta)^2\right].$$

Furthermore, OTDF incorporates a policy regularization term that forces the policy to stay close to the support of the target dataset:

$$\widehat{\mathcal{L}_\pi} = \mathcal{L}_\pi - \beta \times \mathbb{E}_{s\sim\mathcal{D}_{\text{src}}\cup\mathcal{D}_{\text{tar}}} \log \pi^b_{\text{tar}}(\pi(\cdot|s)|s),$$

where $\mathcal{L}_\pi$ is the original policy optimization objective and $\beta$ is the weight coefficient, $\pi^b_{\text{tar}}$ is the behavior policy of the target domain dataset learned with a CVAE. We run the official code[6] for OTDF in our experiments.

## D.2 IMPLEMENTATION DETAILS OF DROCO

In this part, we provide more implementation details of DROCO omitted in the main text.

First, we model the target domain dynamics using a neural network that outputs a Gaussian distribution over the next state: $\widehat{P}_\psi(s'|s,a) = \mathcal{N}(\mu_\psi(s,a), \Sigma_\psi(s,a))$. We learn an ensemble of $N$ dynamics models $\{\widehat{P}_{\psi_i} = \mathcal{N}(\mu_{\psi_i}, \Sigma_{\psi_i})\}_{i=1}^N$, with each model trained independently with maximum likelihood estimation (MLE) on the target domain dataset:

$$\mathcal{L}_{\psi_i} = \mathbb{E}_{(s,a,s')\in\mathcal{D}_{\text{tar}}}\left[\log \widehat{P}_{\psi_i}(s'|s,a)\right]. \tag{28}$$

When we sample from the uncertainty set, we can directly sample from each dynamics model $\mathcal{N}(\mu_{\psi_i}, \Sigma_{\psi_i})$ as the sampling points. We can then compute the value penalty and penalize the Q value of source domain data when leveraging IQL for policy optimization. Specifically, we perform expectile regression to train the V function:

$$\mathcal{L}_V(\eta) = \mathbb{E}_{(s,a)\sim\mathcal{D}_{\text{src}}\cup\mathcal{D}_{\text{tar}}}\left[\mathcal{L}_2^\tau(Q_\theta(s,a) - V_\eta(s))\right],$$

where $\mathcal{L}_2^\tau(u) = |\tau - \mathbb{I}(u < 0)|u^2$ and $\tau \in (0,1)$. For $\tau \approx 1$, $V_\eta$ can capture the in-sample maximal Q (Kostrikov et al., 2021): $V_\eta(s) \approx \max_{a\sim\hat{\mu}(\cdot|s)} Q(s,a)$. We can then practically compute the value penalty as

$$u(s,a,s') = \mathbb{I}\left(s' \sim P_{\text{src}}(\cdot|s,a)\right) \cdot \left(V(s') - \inf_{\{s'_i\}^N \sim \widehat{P}_{\psi_i}(\cdot|s,a)}[V(s'_i)]\right), \tag{29}$$

and the practical Bellman target can be written as

$$\widehat{\mathcal{T}}_{\text{RCB}}Q(s,a) = r(s,a) + \gamma\mathbb{E}_{s'\sim P_\mathcal{M}(\cdot|s,a)}\left[V(s') - \beta \cdot u(s,a,s')\right]. \tag{30}$$

Then, we incorporate Huber loss and have the following Q training loss:

$$\mathcal{L}_Q(\theta) = \mathbb{E}_{\mathcal{D}_{\text{src}}}\left[l_\delta\left(Q_\theta(s,a) - \widehat{\mathcal{T}}_{\text{RCB}}Q_\theta(s,a)\right)\right] + \frac{1}{2}\mathbb{E}_{\mathcal{D}_{\text{tar}}}\left[(Q_\theta(s,a) - \mathcal{T}Q_\theta(s,a))^2\right], \tag{31}$$

where $l_\delta$ is the Huber loss. The final step is policy learning. We follow IQL and utilize exponential advantage-weighted imitation learning to extract the policy:

$$\mathcal{L}_\pi(\phi) = -\mathbb{E}_{\mathcal{D}_{\text{src}}\cup\mathcal{D}_{\text{tar}}}\left[\exp(Q(s,a) - V(s))\log\pi_\phi(a|s)\right].$$

We show the detailed pseudocode of DROCO in Algorithm 1.

---

[6]https://github.com/dmksjfl/OTDF.git

---

**Algorithm 1** Dual-Robust Cross-domain Offline RL (DROCO)

---

1: **Require:** Source domain offline dataset $\mathcal{D}_{\text{src}}$, target domain offline dataset $\mathcal{D}_{\text{tar}}$, mixed offline dataset $\mathcal{D}_{\text{mix}}$

2: **Initialization:** Policy network $\pi_\phi$, value network $V_\eta$, $Q$ network $Q_\theta$, ensemble dynamics model $\widehat{P}_\psi = \{\widehat{P}_{\psi_i}\}_{i=1}^N$, penalty coefficient $\beta$, transition threshold $\delta$ for Huber loss

3: **// Train the ensemble dynamics model**

4: **for** each model gradient step **do**

5:     **for** each ensemble member $\widehat{P}_{\psi_i}$ **do**

6:         Compute loss $\mathcal{L}_{\psi_i} = \mathbb{E}_{(s,a,s') \in \mathcal{D}_{\text{tar}}} \left[ \log \widehat{P}_{\psi_i}(s'|s,a) \right]$

7:         Update $\widehat{P}_{\psi_i}$ using $\mathcal{L}_{\psi_i}$

8:     **end for**

9: **end for**

10: **// TD Learning**

11: **for** each gradient step **do**

12:     Sample $b_{\text{src}} := \{(s,a,r,s')\}$ from $\mathcal{D}_{\text{src}}$

13:     Sample $b_{\text{tar}} := \{(s,a,r,s')\}$ from $\mathcal{D}_{\text{tar}}$

14:     **// Optimize the $V_\beta$ function**

15:     Compute loss $\mathcal{L}_V$:

16:         $\mathcal{L}_V = \mathbb{E}_{(s,a) \sim \mathcal{D}_{\text{src}} \cup \mathcal{D}_{\text{tar}}} [\mathcal{L}_2^\tau (Q_\theta(s,a) - V_\eta(s))]$

17:     Update $V_\eta$ using $\mathcal{L}_V$

18:     **// Compute the value penalty**

19:     compute $u(s,a,s') = \mathbb{I}\left(s' \sim P_{\text{src}}(\cdot|s,a)\right) \cdot \left( V(s') - \inf_{\{s'_i\}^N \sim \widehat{P}_{\psi_i}(\cdot|s,a)} [V(s'_i)] \right)$

20:     **// Optimize the $Q_\theta$ function**

21:     Compute loss $\mathcal{L}_Q$:

22:         $\mathcal{L}_Q = \frac{1}{2} \cdot \mathbb{E}_{(s,a,r,s') \sim \mathcal{D}_{\text{tar}}} \left[ (Q_\theta(s,a) - (r + \gamma V_\eta(s')))^2 \right]$

23:         $+ \frac{1}{2} \cdot \mathbb{E}_{(s,a,r,s') \sim \mathcal{D}_{\text{src}}} [l_\delta (Q_\theta(s,a) - (r + \gamma V_\eta(s') - \beta u(s,a,s')))]$

24:     Update $Q_\theta$ using $\mathcal{L}_Q$

25:     **// Update target network**

26:     Update target network parameters: $\theta' \leftarrow (1 - \mu)\theta + \mu\theta'$

27:     **// Policy Extraction (AWR)**

28:     Compute advantage $A(s,a) = Q_\theta(s,a) - V_\eta(s)$

29:     Optimize policy network $\pi_\eta$ using advantage-weighted regression (AWR):

30:         $\mathcal{L}_\pi = \mathbb{E}_{(s,a) \sim \mathcal{D}_{\text{src}} \cup \mathcal{D}_{\text{tar}}} [\exp(\alpha A(s,a)) \log \pi_\phi(a|s)]$

31: **end for**

---

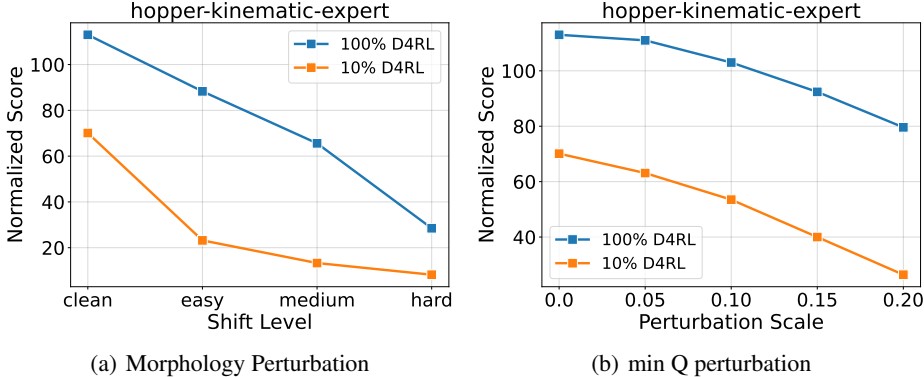

Figure 5: Evaluation results of IGDF under morphology and min Q perturbations with different sizes of target domain data.

# E    EXTENDED EXPERIMENTAL RESULTS

## E.1    EXTENDED RESULTS OF MOTIVATION EXAMPLE

In Section 3, we demonstrate our motivation with a simple example. In this part, we provide more details and results for the motivation example.

The source and target domains are `hopper-kinematic-v2` and `hopper-v2` respectively, with their corresponding datasets being `hopper-kinematic-expert` and `hopper-expert`. Figure 1 in Section 3 demonstrates performance across different target domain data sizes under three test-time kinematic perturbation levels (easy, medium, hard), implemented as in (Lyu et al., 2024b). We further evaluate the trained IGDF under morphology perturbations and min-Q perturbations (with other settings unchanged), presenting results in Figure 5.

The results clearly show that with only 10% D4RL data, IGDF's robustness to dynamics perturbations is significantly weaker compared to using 100% D4RL data. Notably, under easy-level morphology perturbations, IGDF with 10% D4RL data exhibits a 66.9% performance drop, versus only 21.8% degradation with 100% data. These findings, combined with the results in Section 3, validate our motivation that cross-domain offline RL is particularly sensitive to dynamics perturbations when limited target domain data is available, underscoring the need for enhanced test-time robustness.

## E.2    EVALUATION UNDER MORPHOLOGY SHIFTS

In the main text, we present DROCO's evaluation results under kinematic shifts. In this section, we supplement with additional results under morphological shifts, providing a comprehensive assessment of DROCO's train-time robustness against diverse dynamics shifts.

**Experimental Settings.**    The target domain tasks and datasets remain consistent with Section 5.1: the target domain tasks include `halfcheetah-v2`, `hopper-v2`, `walker2d-v2` and `ant-v3`, and the target domain datasets comprise four data qualities (`medium`, `medium-replay`, `medium-expert`, `expert`) for each task. The difference lies in the dynamics shift type in the source domain. We implement morphology shifts as described in Appendix C.3 and collect the corresponding source domain datasets.

**Baselines.** We adopt the same baselines as in Section 5.1: IQL*, CQL*, BOSA, DARA, IGDF and OTDF.

**Results.** We run each baseline and DROCO for 1M steps over 5 random seeds, and present the results with train-time morphology shifts in Table 2. It is clear that DROCO delivers superior performance to baselines. Specifically, DROCO achieves the highest performance in **9** out of 16 tasks. In terms of the total normalized score across all 16 tasks, DROCO attains a remarkable **1166.4**, significantly outperforming the second-best baseline OTDF (1025.1). Combined with the results in

Table 2: **Evaluation results with train-time morphology shifts.** half=halfcheetah, hopp=hopper, walk=walker2d, m=medium, me=medium-expert, mr=medium-replay, e=expert. We report the normalized score evaluated in the target domain, and $\pm$ captures the standard deviation across 5 seeds. We **bold** the highest scores for each task.

| Dataset | IQL$^\star$ | CQL$^\star$ | BOSA | DARA | IGDF | OTDF | DROCO (Ours) |
|---|---|---|---|---|---|---|---|
| half-m | **45.8** | 40.2 | 41.3 | **45.6** | **45.5**±0.1 | 44.3±0.2 | **45.8**±**0.2** |
| half-mr | 26.1 | 21.3 | 27.8 | **28.9** | 24.2±3.3 | 19.7±2.5 | 27.9±4.4 |
| half-me | 63.0 | 54,6 | 44.4 | 59.2 | 61.9± 4.9 | 42.9±3.6 | **70.1**±**5.6** |
| half-e | 65.2 | 66.7 | **78.6** | 55.4 | 56.0±6.2 | 74.2±5.0 | **79.2**±**3.9** |
| hopp-m | **56.4** | 32.8 | 28.7 | 49.5 | 55.5±2.9 | 49.1±2.2 | **56.3**±**1.6** |
| hopp-mr | 51.3 | 37.6 | 40.6 | 53.5 | **54.9**±**5.8** | 24.9±3.4 | 51.6±8.7 |
| hopp-me | 35.8 | 36.6 | 20.2 | 38.2 | 43.3±3.6 | 51.8±3.9 | **82.3**±**4.1** |
| hopp-e | 87.2 | 67.9 | 64.3 | 77.1 | 51.5±2.9 | **113.2**±**5.9** | 92.5±1.2 |
| walk-m | 32.6 | 43.1 | 40.3 | 25.0 | 33.0±2.3 | 40.3±7.1 | **60.1**±**3.4** |
| walk-mr | 9.0 | 2.0 | 2.9 | 6.9 | 9.5±0.4 | 14.1±1.8 | **15.5**±**4.7** |
| walk-me | 27.6 | 22.4 | 46.7 | 42.2 | **75.7**±**11.8** | 66.7±5.3 | **78.9**±**9.4** |
| walk-e | 103.4 | 79.0 | 30.2 | 102.7 | 108.3±6.7 | 103.5±1.9 | 104.5±1.7 |
| ant-m | 89.1 | 57.3 | 36.1 | **96.4** | 91.6±4.4 | 92.5±2.7 | 94.5±2.8 |
| ant-mr | 59.7 | 39.5 | 24.0 | 64.1 | 58.2±7.1 | **69.6**±**8.1** | 66.9±4.9 |
| ant-me | 113.1 | 107.3 | 100.5 | 111.9 | 116.8±3.5 | 107.3±4.4 | **120.3**±**1.5** |
| ant-e | 116.3 | 94.4 | 76.3 | 124.5 | **126.8**±**1.7** | 111.0±2.4 | 120.0±1.3 |
| **Total** | 981.6 | 802.7 | 702.9 | 981.1 | 1012.7 | 1025.1 | **1166.4** |

Section 5.1, these findings conclusively demonstrate DROCO's superiority across different types of dynamics shifts, highlighting its strong train-time robustness against dynamics shifts.

### E.3 EXTENDED EVALUATION UNDER DYNAMICS PERTURBATIONS

In this section, we supplement with more experimental results for evaluating the test-time robustness of DROCO.

We first extend the results in Section 5.2 by incorporating a broader range of datasets. We evaluate the robustness of DROCO against two baselines (IGDF, OTDF) under varying levels of three perturbation types: kinematic, morphology, and min Q perturbations, following the methodology in Section 5.2. Additional experiments are conducted using `hopper-morph-expert`, `walker2d-kinematic-expert`, and `ant-morph-expert` as source domain datasets, with results presented in Figure 6. We can see that DROCO demonstrates superior robustness to all three perturbation types compared to the baselines. For instance, on the `walker2d-kinematic-expert` dataset under min Q perturbations, DROCO exhibits only 23.4% performance degradation (from 106.0 to 81.2) at the highest perturbation level (0.2), substantially lower than IGDF (75.3%) and OTDF (55.9%). This enhanced robustness is consistently observed across all datasets and perturbation types, confirming DROCO's improved test-time robustness against dynamics perturbations.

We further evaluate DROCO's test-time robustness using varying target domain dataset sizes. Experiments are conducted under different levels of min Q perturbations, with target domain sizes set to 100%, 50%, and 10% of the original D4RL datasets. The source domain datasets comprise `hopper-morph-expert`, `walker2d-kinematic-expert`, and `ant-morph-expert`. As shown in Figure 7, all methods demonstrate improved robustness against dynamics perturbations with increasing target domain data size, consistent with our claim in Section 3. Notably, DROCO maintains superior robustness across varying data sizes and perturbation scales compared to IGDF and OTDF, further validating its effectiveness in enhancing test-time robustness.

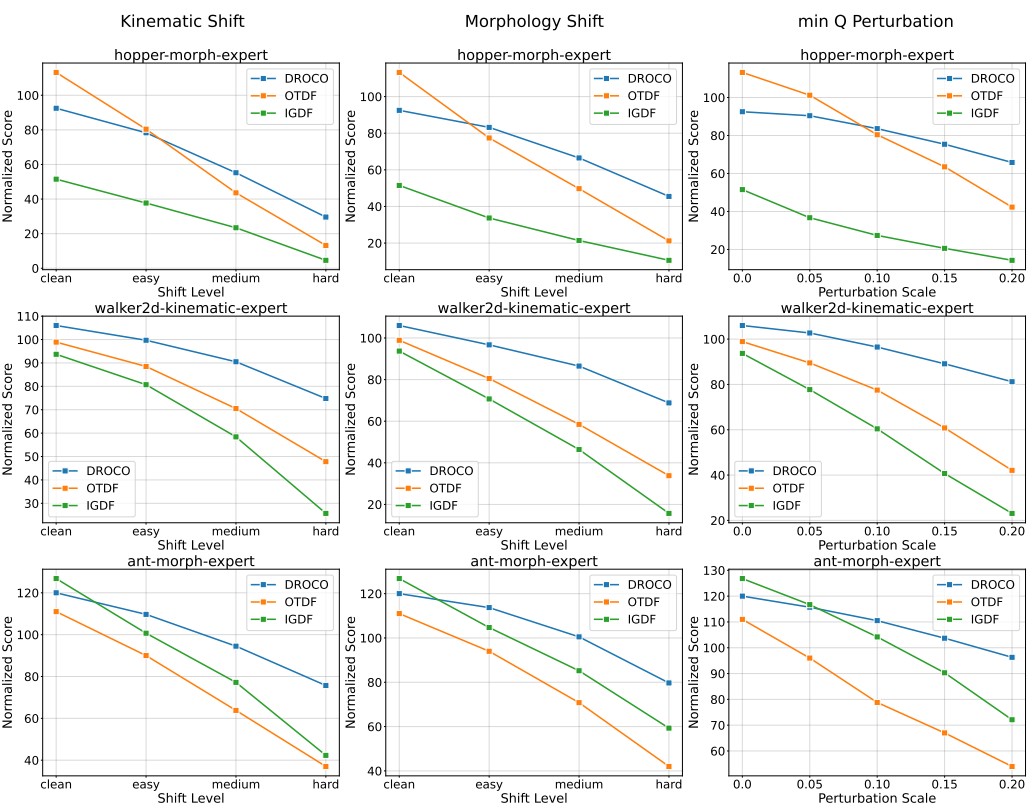

Figure 6: Evaluation results under different types and levels of dynamics perturbations.

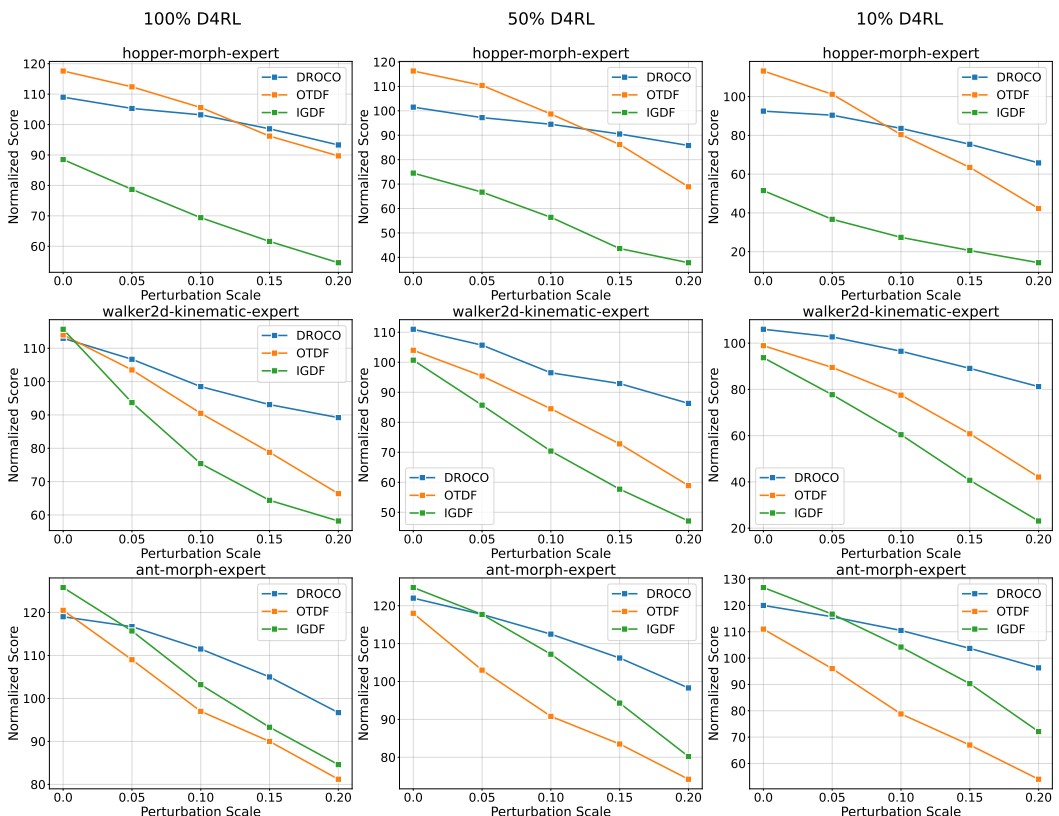

Figure 7: Evaluation results under different perturbation levels and different data sizes.

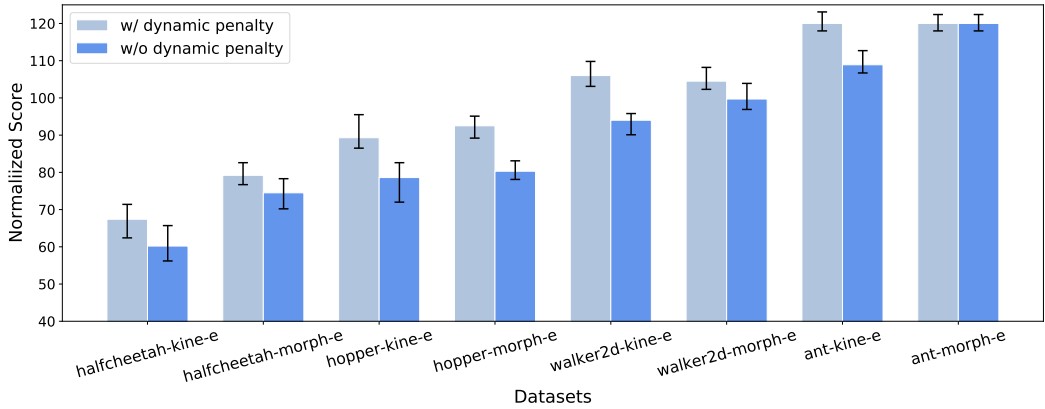

Figure 8: Ablation study on value penalty

### E.4 ABLATION STUDY

We provide supplementary ablation study results that are omitted from the main text. Specifically, we examine the effects of replacing the adaptive value penalty with a fixed value penalty and substituting the Huber loss with the regular $\ell_2$ loss.

**Fixed Value Penalty.** A fixed value penalty corresponds to setting $\beta = 1.0$ across all tasks. Figure 8 compares the performance of DROCO with dynamic versus fixed penalties across eight datasets. The results demonstrate that the dynamic value penalty generally outperforms the fixed penalty ($\beta = 1.0$), except the `ant-morph-expert` dataset where the fixed penalty achieves the highest performance.

We further evaluate the test-time robustness of DROCO under diverse dynamic shifts using both penalty schemes. Following the experimental setup in Appendix E.3, our results in Figure 9 reveal an interesting trade-off: while the fixed value penalty leads to slightly degraded performance, it provides marginally improved robustness against dynamic perturbations. This suggests that setting $\beta$ to a larger value induces a more conservative policy that is less sensitive to dynamic perturbations, albeit at the cost of policy performance.

**Regular $\ell_2$ Loss.** The standard $\ell_2$ loss implements conventional Bellman updates for source domain data without special outlier handling. We evaluate DROCO's performance on 8 `medium-expert` datasets comparing the Huber loss versus the $\ell_2$ loss and present the results in Figure 10. The results show that Huber loss generally produces superior performance, while $\ell_2$ loss achieves marginally better results on `halfcheetah-morph-me` and `walker2d-kine-me` datasets.

We further examine the test-time robustness against dynamic perturbations using both loss functions. Figure 11 reveals that using Huber loss consistently provides stronger robustness across perturbation types, underscoring its critical role in enhancing robustness.

### E.5 EXTENDED PARAMETER STUDY

In the main text, we test the sensitivity of DROCO to the penalty coefficient $\beta$ and the transition threshold $\delta$ on certain datasets. In this section, we present extended results for a more comprehensive analysis.

**Penalty coefficient $\beta$.** $\beta$ controls the intensity of the value penalty. We sweep $\beta$ across $\{0.1, 0.5, 1.0, 1.2\}$ and further conduct experiments on `walker2d-morph-expert` and `ant-morph-expert` datasets. We present the learning curves of the performance and the Q value in Figure 12. We find that $\beta \leq 1.0$ is generally preferred, yielding better performance and Q value convergence, while setting $\beta = 1.2$ would cause value underestimation and inferior performance.

**Transition threshold $\delta$.** $\delta$ determines the transition point from $\ell_2$ loss to $\ell_1$ loss. We vary $\delta$ among $\{5, 10, 30, 50\}$ and conduct experiments on `walker2d-morph-me` and `ant-morph-me`

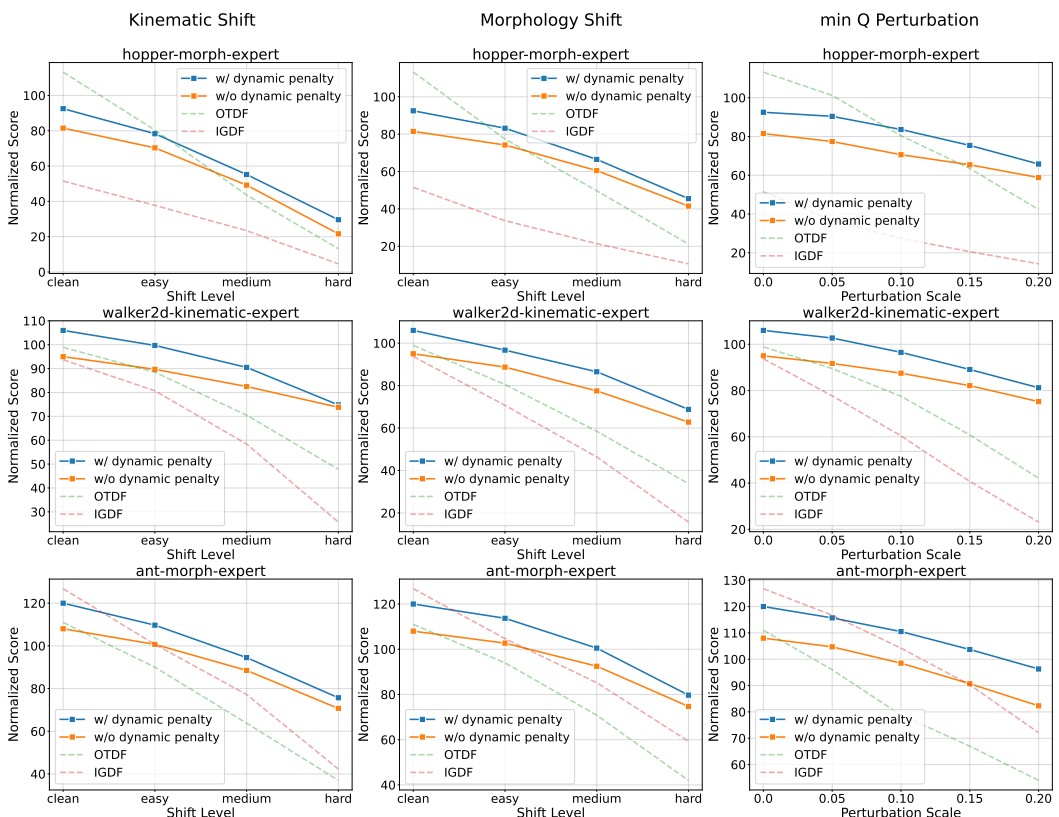

Figure 9: Ablation study on value penalty

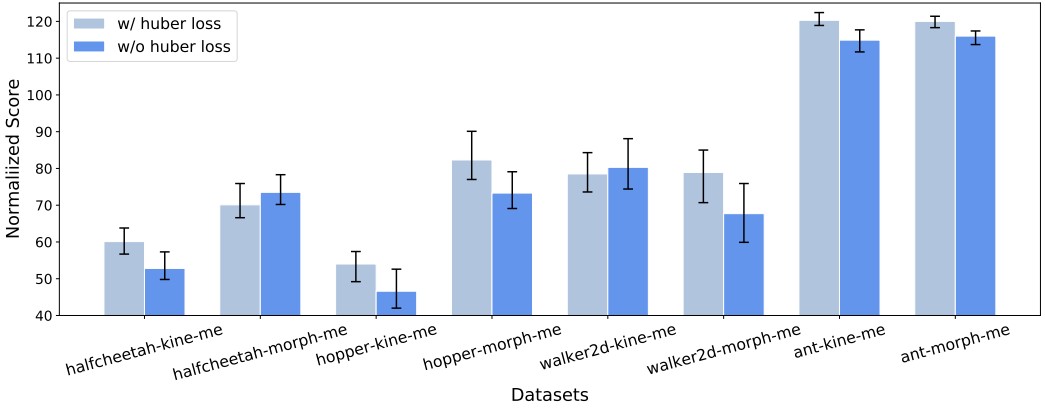

Figure 10: Ablation study on Huber loss

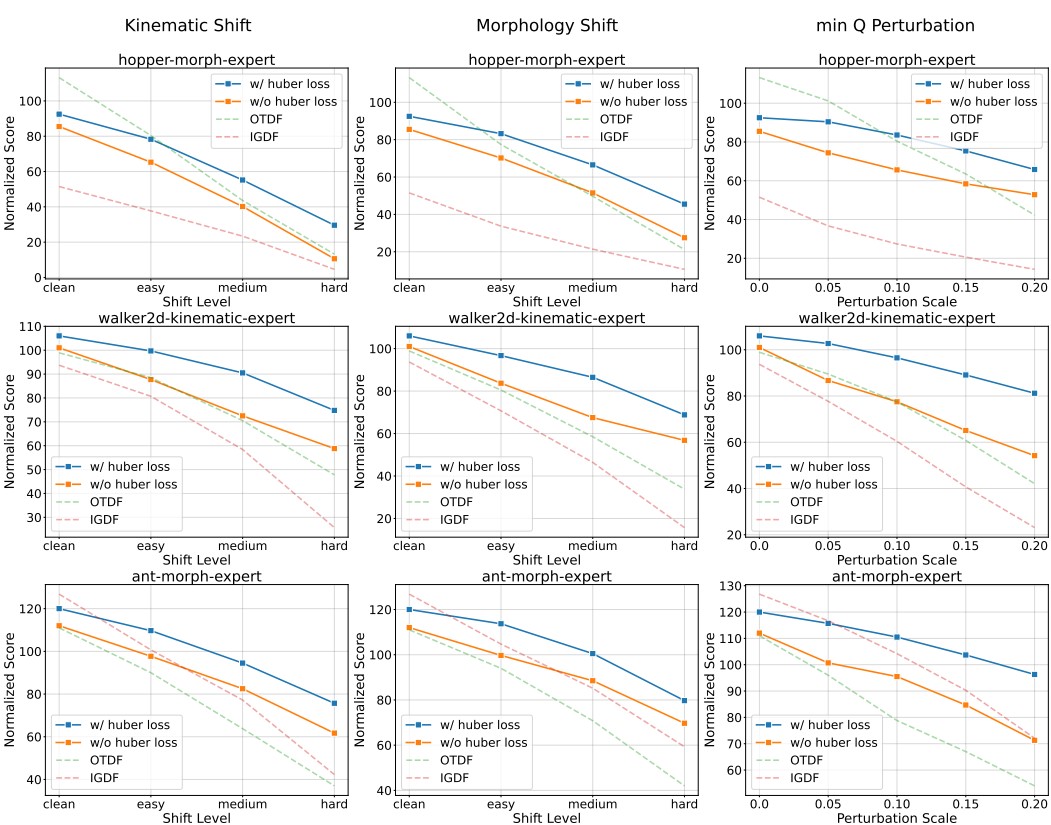

Figure 11: Ablation study on Huber loss

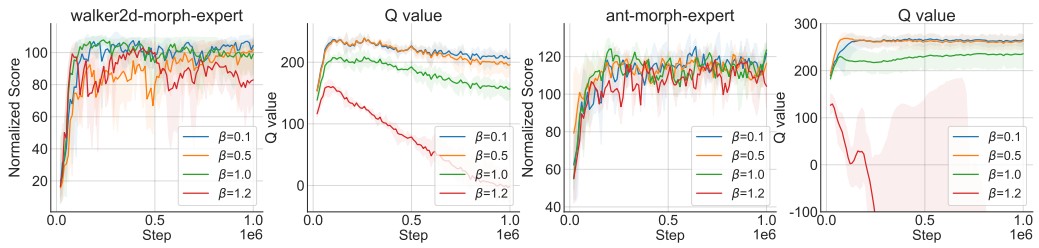

Figure 12: Effect of $\beta$

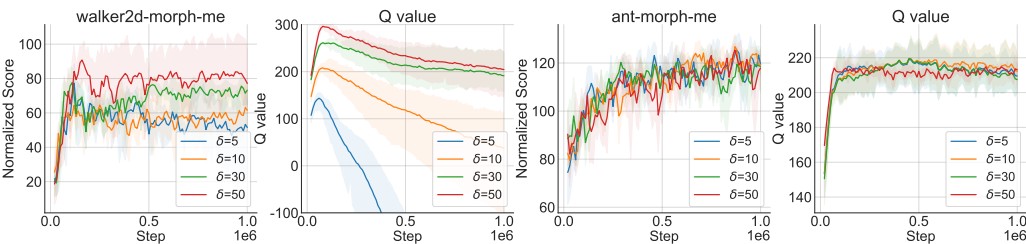

Figure 13: Effect of $\delta$

datasets, with Figure 13 showing the performance and Q value learning curves. Our results demonstrate dataset-dependent sensitivity to $\delta$: while $\delta = 50$ exhibits a satisfying performance and $\delta = 5$ yields suboptimal performance on `walker2d-morph-me`, DROCO is not sensitive to $\delta$ on `ant-morph-me`. Since no single $\delta$ value universally outperforms across all tasks, we specify the $\delta$ values used for each dataset in Appendix F.

**Ensemble size $N$.** We also examine the effect of the dynamics model ensemble size $N$ on training. In DROCO, $N$ represents the sampling number within the uncertainty set. Typically, a larger $N$ corresponds to a smaller sampling error when computing $\widehat{\mathcal{T}}_{\text{RCB}}$. We conduct experiments on various datasets with $N$ across $\{3, 5, 7, 9\}$. The results are presented in Table 3, where we find no distinct difference across different $N$, which means the ensemble size is not a sensitive hyperparameter. Thus, we could use the default value of 7.

Table 3: Effect of $N$

| Dataset | $N = 3$ | $N = 5$ | $N = 7$ | $N = 9$ |
|---|---|---|---|---|
| half-me-kinematic | $58.4 \pm 4.4$ | $62.2 \pm 8.6$ | $60.1 \pm 7.1$ | $57.9 \pm 9.6$ |
| half-me-morph | $65.4 \pm 8.4$ | $71.7 \pm 5.9$ | $70.1 \pm 5.6$ | $74.9 \pm 3.3$ |
| half-e-kinematic | $68.7 \pm 6.8$ | $67.0 \pm 4.7$ | $67.4 \pm 5.8$ | $66.3 \pm 7.2$ |
| half-e-morph | $76.0 \pm 4.1$ | $75.6 \pm 5.1$ | $79.2 \pm 3.9$ | $78.4 \pm 3.0$ |
| hopper-me-kinematic | $51.7 \pm 3.4$ | $54.4 \pm 5.7$ | $54.0 \pm 6.4$ | $52.5 \pm 6.9$ |
| hopper-me-morph | $85.6 \pm 6.7$ | $84.9 \pm 5.5$ | $82.3 \pm 4.1$ | $83.2 \pm 4.0$ |
| hopper-e-kinematic | $88.3 \pm 10.2$ | $87.0 \pm 8.1$ | $89.3 \pm 9.6$ | $86.9 \pm 7.2$ |
| hopper-e-morph | $91.1 \pm 1.0$ | $94.9 \pm 2.2$ | $92.5 \pm 1.2$ | $90.7 \pm 0.8$ |
| Average | $73.2$ | $74.7$ | $74.4$ | $73.9$ |

### E.6 PERFORMANCE COMPARISON UNDER OBSERVATION AND REWARD SHIFTS

In this part, we further examine the generality of DROCO under observation and reward shifts, in addition to dynamics shifts.

**Observation shift.** To simulate the observation shift, we follow the observation corruption setting in (Yang et al., 2024b), and corrupt $30\%$ source domain data by modifying the state of transitions $(s, a, r, s')$ to $\hat{s} = s + \lambda \cdot \text{std}(s), \lambda \sim \text{Uniform}[-1, 1]^{d_s}$. $d_s$ represents the state dimension, and

Table 4: Performance comparison under observation shifts without observation normalization.

| Dataset | IQL | IGDF | OTDF | DROCO |
|---|---|---|---|---|
| half-e-kinematic | 21.7±6.2 | 13.4±2.0 | 28.6±3.4 | **34.7**±5.8 |
| half-e-morph | 43.3±9.5 | 33.8±5.8 | **46.4**±8.3 | 40.8±5.5 |
| half-me-kinematic | 38.9±4.4 | 40.0±4.6 | 37.5±6.2 | **46.3**±9.3 |
| half-me-morph | 37.2±4.1 | **45.3**±5.4 | 33.7±4.0 | 43.4±6.8 |
| hopper-e-kinematic | 34.6±6.4 | 43.5±4.8 | 36.2±7.9 | **48.6**±7.4 |
| hopper-e-morph | 60.6±4.5 | 32.3±4.9 | 53.9±11.5 | **62.9**±8.7 |
| hopper-me-kinematic | 1.4±0.1 | 0.0±0.0 | 16.9±3.1 | **20.3**±7.4 |
| hopper-me-morph | 16.7±2.3 | 22.7±4.0 | 31.4±5.7 | **36.5**±6.8 |
| Average | 31.8 | 28.9 | 35.6 | **41.7** |

Table 5: Performance comparison under observation shifts with observation normalization.

| Dataset | IQL | IGDF | OTDF | DROCO |
|---|---|---|---|---|
| half-e-kinematic | 26.5±4.7 | 21.4±5.0 | 33.8±6.1 | **42.6**±7.1 |
| half-e-morph | 42.7±7.0 | 42.5±5.7 | **51.9**±4.4 | 44.6±7.4 |
| half-me-kinematic | 41.2±5.2 | 35.5±2.9 | 44.3±2.9 | **51.3**±3.6 |
| half-me-morph | 46.4±3.6 | **49.2**±6.0 | 45.6±3.3 | 43.0±2.1 |
| hopper-e-kinematic | 39.6±7.3 | **57.8**±6.2 | 49.3±5.7 | 54.4±9.3 |
| hopper-e-morph | 66.3±6.9 | 38.5±3.7 | 57.7±4.0 | **73.2**±6.2 |
| hopper-me-kinematic | 9.0±1.3 | 2.4±0.1 | 22.2±3.4 | **34.2**±5.6 |
| hopper-me-morph | 16.4±2.4 | 29.7±3.0 | 26.7±1.1 | **46.6**±8.2 |
| Average | 36.0 | 34.6 | 41.4 | **48.7** |

$\text{std}(s)$ is the $d_s$-dimensional standard deviation of all states in the source dataset. Our experiments consist of two parts: **(1)** we directly employ several baselines (IQL, IGDF, OTDF) and DROCO in this observation shift setting without introducing other techniques; **(2)** we introduce the observation normalization technique (Yang et al., 2024b) to baselines and DROCO. Both parts of the experiments are conducted on multiple datasets, with results presented in Table 4 and Table 5. We find that introducing observation shifts would degrade the algorithm's performance, and the observation normalization technique can mitigate performance degradation. In both experimental settings, DROCO demonstrates better performance than baselines on most datasets.

**Reward shift.** To examine the generality of DROCO to reward shifts, we further design a reward shift setting: we randomly select 30% of source transitions $(s, a, r, s')$ and modify the reward $r$ to $\hat{r} \sim \text{Uniform}[-1, 1]$. That is, we completely abandon the reward information and switch to random rewards.

Under this reward shift setting, we conduct experiments on multiple datasets to compare the performance of DROCO with baseline methods (IQL, IGDF, OTDF). The experimental results are reported in Table 6. Surprisingly, we find that reward shift does not significantly affect performance. This observation may be explained by the survival instinct of offline RL (Li et al., 2023), which suggests that offline RL naturally exhibits robustness to misspecified reward.

The results show that DROCO still outperforms other baselines under both reward shift and dynamics shift settings. We attribute the enhanced performance of DROCO under observation and reward shifts to the components of dynamic value penalty and Huber loss which mitigate value estimation error caused by observation and reward shifts. We believe this finding, along with our above results under the observation shift setting, demonstrates the generality of DROCO across observation, reward, and dynamics shift.

Table 6: Performance comparison under reward shifts.

| Dataset | IQL | IGDF | OTDF | DROCO |
|---|---|---|---|---|
| half-e-kinematic | 47.5±4.2 | 45.8±3.0 | **72.2**±3.8 | 66.0±6.3 |
| half-e-morph | 60.7±5.3 | 52.2±3.6 | 70.3±5.8 | **76.4**±4.2 |
| half-me-kinematic | 41.1±3.7 | 55.2±4.4 | 43.6±4.9 | **57.4**±3.6 |
| half-me-morph | 61.7±2.1 | 55.8±4.9 | 39.0±3.3 | **63.9**±2.4 |
| hopper-e-kinematic | 58.8±6.1 | 67.0±5.7 | **95.5**±11.3 | 85.0±8.2 |
| hopper-e-morph | 84.7±5.1 | 46.2±4.8 | **100.3**±6.8 | 91.4±4.5 |
| hopper-me-kinematic | 10.1±1.3 | 8.3±0.7 | 42.6±6.3 | **48.1**±6.4 |
| hopper-me-morph | 34.8±4.3 | 41.1±4.7 | 47.3±5.6 | **78.6**±8.3 |
| Average | 49.9 | 46.5 | 63.9 | **70.9** |

Table 7: Performance comparison under distinct behavior policies between source and target domain datasets.

| Source | Target | IQL | IGDF | OTDF | DROCO |
|---|---|---|---|---|---|
| half-medium | medium | **45.2**±0.1 | **45.2**±0.1 | 42.2±0.1 | **45.3**±0.2 |
| half-medium | expert | 47.5±1.1 | 45.4±1.3 | **58.3**±2.8 | 52.6±4.2 |
| half-expert | medium | 47.1±1.5 | 46.8±2.4 | 51.7±0.4 | **58.5**±0.3 |
| half-expert | expert | 49.7±3.6 | 47.6±2.1 | **79.6**±3.0 | 67.4±5.8 |
| hopper-medium | medium | 48.8±2.1 | 54.3±6.6 | 46.3±3.7 | **55.4**±5.3 |
| hopper-medium | expert | 56.1±4.4 | 61.8±4.4 | 69.3±3.9 | **80.8**±6.2 |
| hopper-expert | medium | 53.6±2.4 | **61.3**±4.7 | 51.4±2.1 | 62.2±4.6 |
| hopper-expert | expert | 62.6±6.9 | 70.1±3.2 | **97.0**±3.3 | 89.3±9.6 |
| walker2d-medium | medium | 48.7±1.9 | 51.8±2.4 | 43.0±2.1 | **70.8**±3.3 |
| walker2d-medium | expert | 71.4±3.7 | 82.5±5.3 | 76.8±4.1 | **94.6**±5.8 |
| walker2d-expert | medium | 55.4±3.1 | 58.6±5.5 | 57.9±2.0 | **83.0**±4.8 |
| walker2d-expert | expert | 90.1±3.2 | 93.7±5.8 | 98.9±2.1 | **106.0**±0.8 |
| ant-medium | medium | 89.9±5.1 | 88.0±4.6 | 86.1±3.7 | **92.7**±6.3 |
| ant-medium | expert | 107.6±1.8 | **112.4**±3.3 | 105.9±2.3 | 110.3±2.0 |
| ant-expert | medium | 93.7±3.5 | 90.2±2.8 | 98.6±4.5 | **100.4**±2.3 |
| ant-expert | expert | 111.0±3.3 | **119.2**±5.6 | 111.6±2.9 | **120.0**±2.1 |
| Average | | 67.4 | 70.6 | 73.4 | **80.6** |

## E.7 PERFORMANCE COMPARISON UNDER DISTINCT SOURCE AND TARGET BEHAVIOR POLICIES

In practice, the behavior policies between the source and target domain datasets could be different. To address this concern, We consider four tasks (halfcheetah, hopper, walker2d, ant) with kinematic shifts. We relax the constraint of identical behavior policies, allowing the source and target datasets to have different qualities (medium or expert). For instance, a medium-quality source dataset may be paired with either a medium- or expert-quality target dataset. All other experimental settings follow Section 5.1, with IQL, IGDF, and OTDF as baselines. The results are presented in Table 7. The results indicate that DROCO maintains its superiority over the baselines even when the source and target behavior policies differ. It achieves the highest average score (80.6) and best performance on 12 out of 16 datasets. These findings demonstrate the effectiveness of DROCO in scenarios with differing behavior policies.

## F HYPERPARAMETER SETUP

In this section, we provide the detailed hyperparameter setup for DROCO in our experiments. In Table 8, we list the network architecture and the training setup of DROCO, as well as the main hyperparameters of IQL, since we utilize IQL for policy optimization. The distinct value of $\beta$ and $\delta$ for each dataset under kinematic shifts and morphology shifts are presented in Table 9 and Table 10.

Table 8: Hyperparameter setup for DROCO

| Hyperparameter | Value |
|---|---|
| **Network** | |
| Actor network | (256, 256) |
| Critic network | (256, 256) |
| Ensemble model network | (400,400,400,400) |
| Ensemble size | 7 |
| Activation function | ReLU (Agarap, 2018) |
| **Training** | |
| Learning rate | $3 \times 10^{-4}$ |
| Optimizer | Adam (Kingma & Ba, 2014) |
| Discount factor | 0.99 |
| Target update rate | $5 \times 10^{-3}$ |
| Source domain batch size | 128 |
| Target domain batch size | 128 |
| Dynamics model batch size | 256 |
| Dynamics model training steps | $1 \times 10^5$ |
| Policy training steps | $1 \times 10^6$ |
| **IQL** | |
| Temperature coefficient | 0.2 |
| Maximum log std | 2 |
| Minimum log std | -20 |
| Inverse temperature parameter $\beta$ | 3.0 |
| Expectile parameter $\tau$ | 0.7 |

Table 9: Detailed hyperparameter setup for DROCO, where the source domain datasets are under **kinematic shifts**.

| Dataset | Value of $\beta$ | Value of $\delta$ |
|---|---|---|
| half-m | 0.1 | 30 |
| half-mr | 0.5 | 50 |
| half-me | 0.5 | 30 |
| half-e | 0.1 | 30 |
| hopp-m | 0.1 | 50 |
| hopp-mr | 0.5 | 50 |
| hopp-me | 1.0 | 30 |
| hopp-e | 0.5 | 30 |
| walk-m | 1.0 | 50 |
| walk-mr | 0.5 | 30 |
| walk-me | 0.5 | 50 |
| walk-e | 0.1 | 10 |
| ant-m | 0.1 | 30 |
| ant-mr | 1.0 | 30 |
| ant-me | 0.1 | 30 |
| ant-e | 1.0 | 30 |

Table 10: Detailed hyperparameter setup for DROCO, where the source domain datasets are under **morphology shifts**.

| Dataset | Value of $\beta$ | Value of $\delta$ |
|---|---|---|
| half-m | 0.1 | 10 |
| half-mr | 0.5 | 50 |
| half-me | 1.2 | 30 |
| half-e | 1.2 | 30 |
| hopp-m | 0.5 | 50 |
| hopp-mr | 0.1 | 50 |
| hopp-me | 0.1 | 10 |
| hopp-e | 0.1 | 10 |
| walk-m | 0.1 | 50 |
| walk-mr | 0.5 | 50 |
| walk-me | 0.1 | 10 |
| walk-e | 0.1 | 10 |
| ant-m | 0.1 | 30 |
| ant-mr | 0.1 | 30 |
| ant-me | 0.1 | 10 |
| ant-e | 1.0 | 30 |

# G    COMPUTE INFRASTRUCTURE

The compute infrastructure we use for all experiments is listed in Table 11.

Table 11: Compute Infrastructure

| **CPU** | **GPU** | **Memory** |
|---------|---------|------------|
| AMD EPYC 7452 | RTX3090$\times$8 | 288GB |

Table 12: Training time comparison between various methods. h=hour(s), m=minute(s).

| IQL$^\star$ | CQL$^\star$ | BOSA | DARA | IGDF | OTDF | DROCO |
|------|-------|------|------|------|------|-------|
| 5h24m | 10h22m | 5h49m | 6h13m | 6h56m | 9h17m | 7h26m |

## H    TIME COST

We list the training time of DROCO and all baselines (IQL$^\star$, CQL$^\star$, BOSA, DARA, IGDF, OTDF) for 1M training steps in Table 12. We note that the additional time cost for DROCO mainly comes from the training of the ensemble dynamics model. However, since we can save the trained dynamics model weights, no retraining is required for subsequent experiments.

## I    BROADER IMPACTS

This paper presents a method aimed at enhancing dual robustness against dynamic shifts in cross-domain offline RL. Our work has potential positive social impacts; for example, it could inspire the development of humanoid robots capable of robust performance in non-stationary environments. Currently, we have not identified any negative impacts of our research.

## J    DECLARATION ON LLM USE

In this work, LLMs are used solely for grammar polishing of an early draft and are excluded from core aspects of the research, such as method conception, theoretical proof, and experimental work.

