# OpenReview forum: "Dual-Robust Cross-Domain Offline Reinforcement Learning Against Dynamics Shifts"
_ICLR.cc/2026/Conference — ICLR 2026 Poster_

### Official Review · Reviewer_k93k · 2025-10-27

**Soundness:** 3
**Presentation:** 3
**Contribution:** 2
**Rating:** 6
**Confidence:** 3

**Summary:**

This paper investigates dual robustness in cross-domain offline RL, addressing both train-time robustness against source-target dynamics mismatch and test-time robustness against deployment-time dynamics perturbations. The authors introduce a novel Robust Cross-domain Bellman (RCB) operator with theoretical guarantees, and develop the practical DROCO algorithm using ensemble dynamics models, dynamic value penalty, and Huber loss. The experimental results across 32 scenarios demonstrate that DROCO achieves good performance.

The authors employ dynamics perturbations of large magnitude, which may diverge from realistic scenarios. More critically, to put the theory into practice, the algorithm adopts a practical scheme that deviates from its core theoretical assumptions: using an ensemble of dynamics models trained on limited target data to approximate uncertainty. To compensate for the potential value estimation errors caused by this deviation, DROCO further introduces a dynamic value penalty and the Huber loss.

Although the method demonstrates performance surpassing baselines and enhanced robustness on specific Mujoco tasks, its effectiveness is highly dependent on a set of sensitive hyperparameters that are difficult to tune effectively in a purely offline setting. Furthermore, the experimental scope is confined to locomotion tasks and has not been validated on more challenging benchmarks, which leaves the generalizability and practical value of its claimed "dual robustness" open to question.

**Strengths:**

The problem formulation is both novel and practically important. While existing cross-domain offline RL methods focus exclusively on train-time robustness, this work is the first to systematically study both train-time and test-time robustness together. The motivation is compelling, with Figure 1 clearly demonstrating that policies trained with limited target domain data are highly vulnerable to test-time dynamics perturbations. This observation reveals a critical gap in current approaches that assume deployment environments will match the target domain exactly.

The theoretical contributions are well-developed and rigorous. The RCB operator elegantly handles dual robustness by applying robust Bellman updates only to source domain data while using standard updates for target data.

**Weaknesses:**

My primary concern is the insufficient analysis of generalization. Moreover, the experiments are confined entirely to MuJoCo tasks. Maybe authors can consider more experiment for validation.

The paper's own sensitivity analysis (Section 5.3) showed that the optimal values for β and δ vary significantly across different tasks and datasets. In a real-world offline scenario, it is nearly impossible to tune these parameters to their optimal values for a new task due to the inability to validate against the target environment. The method's claim to practicality is diminished if its strong performance relies on meticulous, per-task tuning that cannot be replicated in practice.

**Questions:**

See Weaknesses

---

> ### Author Response · Authors · 2025-11-22
> **Reply to Reviewer k93k**
>
> We thank the reviewer for the insightful review. We appreciate that the reviewer thinks our work is novel and practically important. We present our responses to the major concerns below. We hope these responses could address the reviewer's concerns.
>
> ## Concern 1: More experiments in addition to MuJoCo tasks
>
>
> As demonstrated in subsection 1 in Appendix A, existing literature [1] indicates that tasks like Antmaze and Adroit are highly challenging for cross-domain RL, and enabling cross-domain RL to succeed in such challenging tasks remains an open problem and falls beyond the scope of this work. Moreover, the focus on locomotion tasks in recent cross-domain RL studies [4,5,6] has made this a standard setting for evaluating cross-domain RL algorithms.
>
> However, we appreciate the suggestion of testing DROCO on more tasks. Given that Antmaze and Adroit remain too challenging, we turn to the DeepMind Control Suite (DMC) and select three tasks for evaluation: Fish-swim, Swimmer-15, and Humanoid-run. The numbers in parentheses denote the state and action dimensions (dim(S), dim(A)) for each task: Fish-swim (26, 5), Swimmer-15 (34, 14), Humanoid-run (54, 21).
>
> We exclude tasks like Pendulum and Acrobot due to their low state/action dimensions, as well as Cheetah, Hopper and Walker, which are similar to the MuJoCo tasks already evaluated in our main experiments. Since DMC only provides online environments, we collect medium and expert offline datasets by collecting transitions using SAC policies trained to medium and expert levels for each task. The target dataset for each task contains approximately 100,000 transitions.
>
> For the source domain, we introduce kinematic and morphology shifts similar to our main experiments by modifying the .xml files governing the task's dynamics. We collect medium and expert source datasets for each task, totaling around 1,000,000 transitions. We then train baseline methods (IQL, IGDF, OTDF) and DROCO on these datasets and evaluate policy performance in the target domain. We note that we do not bother tuning the hyperparameters $\beta$ and $\delta$ for our DMC experiments. We fix $\beta=0.1$ and $\delta=50$ across all datasets.
>
> **The results shown below indicate that DROCO still outperforms other baselines on 7/12 datasets and achieves the highest average score**. We believe these results on DMC tasks could serve as evidence that DROCO could be generalized to tasks other than MuJoCo tasks.
>
> ||IQL|IGDF|OTDF|DROCO|
> |-|-|-|-|-|
> |Fish-m-kinematic|333$\pm$56|392$\pm$102|483$\pm$55|**523$\pm$74**|
> |Fish-m-morph|421$\pm$49|468$\pm$84|425$\pm$30|**487$\pm$63**|
> |Fish-e-kinematic|765$\pm$42|783$\pm$6|**815$\pm$24**|800$\pm$10|
> |Fish-e-morph|680$\pm$46|733$\pm$28|**795$\pm$10**|778$\pm$4|
> |Swimmer-m-kinematic|366$\pm$16|358$\pm$49|305$\pm$82|**400$\pm$35**|
> |Swimmer-m-morph|405$\pm$32|381$\pm$43|340$\pm$65|**428$\pm$17**|
> |Swimmer-e-kinematic|545$\pm$50|**620$\pm$14**|496$\pm$112|584$\pm$25|
> |Swimmer-e-morph|575$\pm$25|600$\pm$36|562$\pm$59|**616$\pm$15**|
> |Humanoid-m-kinematic|340$\pm$22|318$\pm$77|**380$\pm$32**|357$\pm$66|
> |Humanoid-m-morph|322$\pm$14|304$\pm$48|373$\pm$29|**405$\pm$45**|
> |Humanoid-e-kinematic|645$\pm$38|**674$\pm$40**|607$\pm$55|630$\pm$15|
> |Humanoid-e-morph|606$\pm$30|610$\pm$49|626$\pm$85|**695$\pm$31**|
> |Average|500.3|483.4|508.1|**567.8**|
>
>
> ## Concern 2: On the hyperparameter tuning
>
>
> As we demonstrate in the parameter sensitivity analysis (Section 5.3), although the optimal $\beta$ and $\delta$ vary across different tasks, there are common patterns for these tasks. we find $\beta\leq1.0$ and a larger $\delta$ work for most tasks. Therefore, for a new task, we could first try $\beta\leq 1.0$ and $\delta=30$ or $\delta=50$. This could be a guideline for hyperparameter selection.
>
> Moreover, even without access to the online environment, hyperparameters can be selected using a simple heuristic during offline training. For example, we could select hyperparameters according to $\min(Q_{avg}+Q_{var})$, based on the principle that the value function should be regularized and stable. This selection method has been used in prior offline RL literature such as COMBO [2] and RAMBO [3].
>
> [1] ODRL: A Benchmark for Off-Dynamics Reinforcement Learning. NeurIPS 2024
>
> [2] Conservative Offline Model-Based Policy Optimization. NeurIPS 2021
>
> [3] RAMBO-RL: Robust Adversarial Model-Based Offline Reinforcement Learning. NeurIPS 2022
>
> [4] Cross-domain Offline Policy Adaptation with Optimal Transport and Dataset Constraint. ICLR 2025
>
> [5] Composite Flow Matching for Reinforcement Learning with Shifted-Dynamics Data. NeurIPS 2025
>
> [6] Contrastive Representation for Data Filtering in Cross-Domain Offline Reinforcement Learning. ICML 2024

---

> > ### Comment · Reviewer_k93k · 2025-11-28
> > **Comment**
> >
> > Thanks for the response. I keep my positive score.

---

> > > ### Author Response · Authors · 2025-11-28
> > >
> > > We are glad that the concerns are addressed. We thank the reviewer for the positive rating and for the devoted time and effort in reviewing our paper.

---

### Official Review · Reviewer_XtEu · 2025-10-27

**Soundness:** 3
**Presentation:** 3
**Contribution:** 3
**Rating:** 6
**Confidence:** 3

**Summary:**

This paper enhances offline RL by introducing two types of robustness: train-time robustness and test-time robustness. Train-time robustness addresses the dynamic shift between the source-domain dataset and the target-domain dataset, while test-time robustness focuses on the shift from the target dynamics to the deployment dynamics. Considering that (1) deployment dynamics may change in real-world settings and (2) existing methods often fail to generalize well under such changes, I believe this paper makes a valuable contribution.

The proposed method centers around a novel Robust Cross-domain Bellman (RCB) operator, which integrates two types of Bellman operations: (1) the standard Bellman operator for training on the target-domain dataset and (2) a “robust” Bellman operator for policy evaluation on the source-domain data. The theoretical analysis, in my view, can be derived from classic results in robust reinforcement learning.

The theoretical justification mainly concerns the convergence (contraction) property of the proposed RCB operator. I believe that the conclusions for both the idealized and practical cases can be obtained with relatively minor modifications to existing robust RL proofs.

The empirical results (Table 1, on MuJoCo tasks) adequately demonstrate the advantages of the proposed algorithm.

**Strengths:**

- The theoretical justification is solid, covering both the idealized case (Proposition 4.1) and the practical case (Proposition 4.3).

- The motivation and analysis for both train-time and test-time robustness  (Proposition 4.4 and 4.5) are meaningful and potentially impactful, although their direct relevance to practitioners might be limited.

- The empirical evaluation is convincing. The chosen baselines are sota methods for offline RL and cross-domain offline RL, yet the proposed algorithm (DROCO) achieves significant improvements over them.

**Weaknesses:**

I do not see any major weaknesses worth highlighting.

**Questions:**

Q1. Lipschitz Q-function assumption:
Why do you cite recent studies to justify the Lipschitz continuity assumption? If I am not mistaken, this assumption is a standard one in Q-learning and can be found in many textbooks.

Q2. Test-time robustness:
It is somewhat difficult for me to see the benefits of DROCO regarding test-time robustness. Could you please elaborate on this aspect explicitly in the Experiments section?

---

> ### Author Response · Authors · 2025-11-22
> **Reply to Reviewer XtEu**
>
> We thank the reviewer for the thoughtful review and for acknowledging the contributions of our work. We present responses to the major concerns below. We hope our responses could address the reviewer's concerns.
>
> ## Concern 1: The justification of Lipschitz continuity assumption
>
> The Lipschitz continuity assumption is a standard and common one in RL, and as such, we cite recent studies to help readers who wish to find the relevant literature on this topic.
>
>
> ## Concern 2: Adding elaboration on the test-time robustness
>
> We appreciate this suggestion. Test-time robustness can be measured by the degree of performance degradation in the face of dynamic perturbations, compared to the clean environment. In Section 5.2, we evaluate DROCO and other baselines by exposing them to three types of dynamic shifts (kinematic, morphological, and min Q perturbations) at various levels. We consider DROCO to exhibit better test-time robustness if it demonstrates less performance degradation than the baselines under the same shift severity. For instance, under easy-level kinematic shifts, DROCO's performance degrades by only 19.3%, whereas IGDF and OTDF suffer degradations exceeding 50%. This result demonstrates DROCO's superior test-time robustness. We have elaborated on this point explicitly in the revised Section 5.2.

---

> > ### Author Response · Authors · 2025-11-28
> >
> > Dear Reviewer XtEu, thanks for your thoughtful review. As the author-reviewer discussion period is near its end, we wonder if our rebuttal addresses your concerns. Please let us know if any further clarifications or discussions are needed!

---

### Official Review · Reviewer_z4fr · 2025-10-28

**Soundness:** 3
**Presentation:** 3
**Contribution:** 2
**Rating:** 4
**Confidence:** 4

**Summary:**

The paper studies dual robustness in cross-domain offline RL. It considers train time robustness to distribution shift between source and target, and test-time robustness to perturbations around the target dynamics. It proposes a Robust Cross-domain Bellman (RCB) operator that performs robust backups (min over uncertainty set) on source data and standard backups on target data. Experiments are performed on D4RL MuJoCo tasks to show performance gains and test time stability.

**Strengths:**

+ The goal of achieving both train time and test time robustness in offline RL is well motivated.
+ The RCB operator that separates robust and standard updates is simple especially with the duality result which simplifies the uncertainty set of distributions to one over states.
+ The paper gives good empirical results with RL benchmarks showing that the approach outperforms baselines under moderate dynamics shifts.

**Weaknesses:**

- The robust Bellman backups and the derived contraction properties are standard. As far as I understand, the main idea is to split robust and standard updates, which is conceptually incremental. The paper seems conceptually incremental for ICLR.
- The setup is restrictive as only dynamics shift is modeled. Typically, there is shift in reward, observation, state/action spaces, etc.
- The theoretical results largely follow directly from known robust RL results. For example Prop 4.1 showing the contraction is immediate for discounted robust Bellman operators. The train time conservatism and lower bound properties (e.g., Prop 4.4) are classic robust RL analyses. Even the test time guarantee of Prop 4.5 arguing that the performance is better than the worst case value when the true perturbation lies inside the set is by construction and standard in DRO.
- The framework is restricted to Wasserstein distance without extensions to TV/MMD or other divergences.
- It is unclear how to choose or tune the uncertainty radius. Also if the domain shift is large, the approach is likely going to be over conservative given the requirement that the target lies in the uncertainty set.
- I did not quite understand the value penalty and the Huber loss parts as they are treated superficially without sufficient depth.

**Questions:**

What is the precise theoretical and/or algorithmic novelty beyond the split backup? Any coverage aware or calibration guarantees that are new? The paper needs better positioning in relation to vast literature on robust offline RL/DRO.

Can your approach extend to TV/MMD balls (even approximately)? What breaks or becomes intractable?

Any results for reward or state shift to demonstrate generality?

How does performance scale with the amount of target data and epsilon? Where does RCB become too conservative?

The source and target behavior policies are often different. Can you analyze this setting?

---

> ### Author Response · Authors · 2025-11-22
> **Reply to Reviewer z4fr (Part 1)**
>
> We thank the reviewer for the thoughtful review and for acknowledging the strengths of our work. We provide responses to the major concerns below. We hope our responses could address the reviewer's concerns.
>
> ## Concern 1: Positioning and novelty of DROCO
>
> - **The positioning of DROCO.** We respectfully clarify that this work is positioned within the setting of **cross-domain offline RL, rather than single-domain robust RL**. Our core contributions lie in **the insight that enhancing dual robustness in cross-domain RL from a robsut RL perspective, instead of advancing robust RL itself**. Our goal is not to introduce new theoretical analysis, but to adapt robust Bellman backups to achieve dual robustness in the cross-domain offline setting. While this work does not focus on novel theoretical analysis, we believe it can inspire future research within the cross-domain RL community. Moreover, **DROCO offers a practical algorithm to address both train-time and test-time robustness, which existing robust RL/DRO have not yet explored**.
>
> - **The novelty of DROCO.** The novelty of DROCO lies in **its core insight, i.e., the emphasis on dual robustness in cross-domain offline RL, and the introduction of the RCB operator for its implementation**. Although the underlying theoretical analysis applies some tools in robust RL, our work rigorously integrates them into cross-domain RL and establishes that dual robustness can be achieved with our RCB operator. Our objective is to provide a comprehensive and practical solution for cross-domain RL rather than to pursue a theoretical novelty for single-domain robust RL. Algorithmically, **DROCO's novelties beyond split backup include**: **(1)** the design of a practical RCB operator that transforms dynamics uncertainty into state uncertainty; **(2)** the use of an ensemble dynamics model for flexible sampling; and **(3)** the incorporation of a dynamic value penalty and Huber loss to address value estimation error. These components make DROCO practical to fulfill dual robustness.
>
> We hope our response could help the reviewer better position our work.
>
> ## Concern 2: DROCO does not consider other shift types such as reward, observation, state/action spaces
>
> **Our initial focus on dynamics shift aligns with the standard setting in prior cross-domain RL research**. **We appreciate the reviewer’s suggestion and have conducted additional experiments under observation shift and reward shift**.
>
> **Observation shift.** To simulate the observation shift, we follow the observation corruption setting in [1], and corrupt 30\% source domain data by modifying the state of transitions $(s,a,r,s^\prime)$ to $\hat{s}=s+\lambda\cdot\mathrm{std}(s), \lambda\sim\mathrm{Uniform}[-1,1]^{d_s}$. $d_s$ represents the state dimension, and $\mathrm{std}(s)$ is the $d_s$-dimensional standard deviation of all states in the source dataset. Our experiments consist of two parts: **(1)** we directly employ several baselines (IQL, IGDF, OTDF) and DROCO in this observation shift setting without introducing other techniques; **(2)** we introduce the observation normalization technique [1] to baselines and DROCO. Both parts of the experiments are conducted on multiple datasets, with results presented below. We find that introducing observation shifts would degrade the algorithm's performance, and the observation normalization technique can mitigate performance degradation. In both experimental settings, **DROCO demonstrates better performance than baselines on most datasets**.
>
> |w/o observation normalization|IQL|IGDF|OTDF|DROCO|
> |-|-|-|-|-|
> |half-e-kinematic| 21.7$\pm$6.2 |13.4$\pm$2.0 |28.6$\pm$3.4 |**34.7**$\pm$5.8 |
> |half-e-morph| 43.3$\pm$9.5 |33.8$\pm$5.8 | **46.4**$\pm$8.3| 40.8$\pm$5.5 |
> |half-me-kinematic| 38.9$\pm$4.4 |40.0$\pm$4.6 | 37.5$\pm$6.2 | **46.3**$\pm$9.3 |
> |half-me-morph| 37.2$\pm$4.1 |**45.3**$\pm$5.4|33.7$\pm$4.0 |43.4$\pm$6.8 |
> |hopper-e-kinematic|34.6$\pm$6.4 |43.5$\pm$4.8 | 36.2$\pm$7.9| **48.6**$\pm$7.4 |
> |hopper-e-morph| 60.6$\pm$4.5 | 32.3$\pm$4.9|53.9$\pm$11.5 |**62.9**$\pm$8.7|
> |hopper-me-kinematic| 1.4$\pm$0.1 |0.0$\pm$0.0 |16.9$\pm$3.1 |**20.3**$\pm$7.4|
> |hopper-me-morph|16.7$\pm$2.3 |22.7$\pm$4.0 |31.4$\pm$5.7 |**36.5**$\pm$6.8|
> |Average| 31.8 |28.9| 35.6 | **41.7** |
>
> |w/ observation normalization|IQL|IGDF|OTDF|DROCO|
> |-|-|-|-|-|
> |half-e-kinematic| 26.5$\pm$4.7 | 21.4$\pm$5.0 | 33.8$\pm$6.1 |**42.6**$\pm$7.1|
> |half-e-morph| 42.7$\pm$7.0 |42.5$\pm$5.7|**51.9**$\pm$4.4|44.6$\pm$7.4|
> |half-me-kinematic| 41.2$\pm$5.2 |35.5$\pm$2.9|44.3$\pm$2.9 |**51.3$\pm$3.6**|
> |half-me-morph| 46.4$\pm$3.6 |**49.2$\pm$6.0**| 45.6$\pm$3.3|43.0$\pm$2.1|
> |hopper-e-kinematic|39.6$\pm$7.3|**57.8$\pm$6.2**| 49.3$\pm$5.7 |54.4$\pm$9.3|
> |hopper-e-morph|66.3$\pm$6.9 |38.5$\pm$3.7|57.7$\pm$4.0 |**73.2**$\pm$6.2|
> |hopper-me-kinematic|9.0$\pm$1.3 | 2.4$\pm$0.1| 22.2$\pm$3.4 |**34.2**$\pm$5.6|
> |hopper-me-morph|16.4$\pm$2.4 |29.7$\pm$3.0|26.7$\pm$1.1 |**46.6**$\pm$8.2 |
> |Average | 36.0 |34.6 |41.4 |**48.7** |

---

> ### Author Response · Authors · 2025-11-22
> **Reply to Reviewer z4fr (Part 2)**
>
> **Reward shift.** To examine the generality of DROCO to reward shifts, we further design a reward shift setting: we randomly select 30\% of source transitions $(s,a,r,s^\prime)$ and modify the reward $r$ to $\hat{r}\sim \mathrm{Uniform}[-1,1]$. That is, we completely abandon the reward information and switch to random rewards.
>
> Under this reward shift setting, we conduct experiments on multiple datasets to compare the performance of DROCO with baseline methods (IQL, IGDF, OTDF). The experimental results are reported below. Surprisingly, we find that reward shift does not significantly affect performance. This observation may be explained by the survival instinct of offline RL [2], which suggests that offline RL naturally exhibits robustness to misspecified reward.
>
> **The results show that DROCO still outperforms other baselines under both reward shift and dynamics shift settings**. We attribute the enhanced performance of DROCO under observation and reward shifts to the components of dynamic value penalty and Huber loss which mitigate value estimation error caused by observation and reward shifts. We believe this finding, along with our above results under the observation shift setting, demonstrates **the generality of DROCO across observation, reward, and dynamics shift**. The above results have been incorporated in Appendix E.6 in our revision
>
> |reward shift|IQL|IGDF|OTDF|DROCO|
> |-|-|-|-|-|
> |half-e-kinematic|47.5$\pm$4.2|45.8$\pm$3.0|**72.2$\pm$3.8**|66.0$\pm$6.3|
> |half-e-morph|60.7$\pm$5.3|52.2$\pm$3.6|70.3$\pm$5.8|**76.4$\pm$4.2**|
> |half-me-kinematic|41.1$\pm$3.7|55.2$\pm$4.4|43.6$\pm$4.9|**57.4$\pm$3.6**|
> |half-me-morph|61.7$\pm$2.1|55.8$\pm$4.9|39.0$\pm$3.3|**63.9$\pm$2.4**|
> |hopper-e-kinematic|58.8$\pm$6.1|67.0$\pm$5.7|**95.5$\pm$11.3**|85.0$\pm$8.2|
> |hopper-e-morph|84.7$\pm$5.1|46.2$\pm$4.8|**100.3$\pm$6.8**|91.4$\pm$4.5|
> |hopper-me-kinematic|10.1$\pm$1.3|8.3$\pm$0.7|42.6$\pm$6.3|**48.1$\pm$6.4**|
> |hopper-me-morph|34.8$\pm$4.3|41.1$\pm$4.7|47.3$\pm$5.6|**78.6$\pm$8.3**|
> |Average| 49.9 |46.5| 63.9 | **70.9** |
>
> **State-action space.** DROCO adopts **the standard setting of recent studies (IGDF, OTDF)**, which assumes an identical state-action space across source and target domains. However, our method has the potential to be extended to handle distinct state-action representations by incorporating inter-domain mapping techniques, such as dynamics cycle consistency [6]. We leave the extension of DROCO to such heterogeneous state-action spaces as future work.
>
> ## Concern 3: The theoretical results follow Robust RL results
>
> The contraction property and dual robustness analysis build upon some tools in robust RL. However, as discussed in the response to Concern 1, the key contribution of our theoretical analysis is not to advance robust RL theory itself, but to identify and formalize the critical role of dual robustness for cross-domain offline RL, an aspect overlooked by prior work in this field. We believe this insight provides a valuable foundation for future research.
>
> ## Concern 4: Can DROCO be extended to other divergences?
>
> It is known that if choosing the **discrete** metric $d(x,y)=\mathbb{I}(x\neq y)$ as the cost function in Wasserstein distance, then $\\mathcal{W} _ 1(p,q)=D _ {TV}(p,q)$ [3]. **In this case, our method can be seamlessly extended to the TV balls**.
>
> However, this equivalence does not hold for other divergences or cost functions. Therefore, DROCO cannot be directly applied to them, as it would require deriving a new dual reformulation for each specific divergence (e.g., $f$-divergence). **The dual reformulations for these divergences are often complex and involve solving complicated constrained optimization problems [4, 5], making them impractical to implement**.
>
> This is why we adopt the Wasserstein uncertainty set: **it admits an elegant closed-form dual reformulation (Proposition 4.2), which allows us to convert the dynamics uncertainty set into a simple state uncertainty set**. This property is critical for our practical implementation.
>
> ## Concern 5: It is unclear how to choose the uncertainty radius, and DROCO could be over-conservative if the dynamics shift is large
>
> This issue is precisely what our Practical Algorithm part (Section 4.3) aims to address. Our solution to the over-conservatism issue is to determine the uncertainty set via ensemble dynamics modeling instead of fixing $\\epsilon$. Specifically, we only require the uncertainty set around $s^\\prime _ {tar}$ instead of $s^\\prime _ {src}$ to alleviate the unnecessary conservatism. To achieve this, we train an ensemble dynamics model in the target domain and treat each ensemble member's prediction as a sample from the uncertainty set. In this way, we alleviate the unnecessary conservatism when the source and target dynamics deviate far, and we do not need to manually choose the uncertainty set radius $\epsilon$, since $\epsilon$ is implicitly determined by the variability in the ensemble predictions.

---

> ### Author Response · Authors · 2025-11-22
> **Reply to Reviewer z4fr (Part 3)**
>
> ## Concern 6: On the value penalty and Huber loss
>
> We respectfully argue that **the design of the dynamic value penalty and Huber loss is well-motivated** and is not treated superficially. Below, we provide a clearer explanation of their motivation and implementation.
>
> **Motivation.** Recall that we treat the ensemble predictions as the samples from the uncertainty set. However, since the ensemble size is typically limited (such as 7), the ensemble prediction set is unlikely to cover the support of $P_{tar}$ as required by Proposition 4.4, such that Q-value estimation error (overestimation or underestimation) might occur. Therefore, we further introduce two techniques to DROCO: the dynamic value penalty and Huber loss to deal with the value estimation error.
>
> **Implementation.** Then we explain how the dynamic value penalty and Huber loss tackle value estimation error. By introducing a value penalty term $u(s,a,s^\\prime)$ (Equation (6)), we could unify the RCB operator under source and target dynamics (Equation (7)), and use a tunable $\beta$ to scale the penalty. If we set $\beta=1.0$, then we recover the original RCB operator for updates. If we set $\beta>1.0$, then the penalty is increased to mitigate value overestimation. If we set $\beta<1.0$, then the penalty is decreased to mitigate value underestimation. And the Huber loss is a well-established technique for enhancing the robustness against outliers. If the value estimation error is large, then the $\ell_2$ loss would turn to $\ell_1$ loss to mitigate the value estimation error. By introducing these two techniques for implementation, we enable flexibility to deal with the underlying value estimation error.
>
> We hope our explanation could help the reviewer better understand the motivation and the design of this part.

---

> ### Author Response · Authors · 2025-11-22
> **Reply to Reviewer z4fr (Part 4)**
>
> ## Concern 7:  How does performance scale with the amount of target data and epsilon? When does RCB become too conservative?
>
> **How does performance scale with the amount of target data and epsilon?** As addressed in our response to Concern 5, $\epsilon$ is implicitly determined by the ensemble dynamics model, eliminating the need for manual selection. Therefore, we focus here on evaluating how DROCO's performance scales with the amount of target data. In our original experiments, we use 10% of the D4RL dataset as the target domain. We vary the target dataset size across {5%, 10%, 20%, 50%} of the D4RL data and compare the performance of DROCO with baseline methods (IQL, IGDF, OTDF) under kinematic shifts. All other experimental settings remain consistent with Section 5.1. The results are presented below.
>
> |Dataset (5\%)|IQL|IGDF|OTDF|DROCO|
> |-|-|-|-|-|
> |half-m | **43.4$\pm$0.2** | 42.6$\pm$0.1 |40.9$\pm$0.3 | 40.0$\pm$0.1 |
> |half-mr |17.3$\pm$1.6 | 19.8$\pm$2.0 |11.3$\pm$2.4 |**21.6$\pm$2.8** |
> |half-me |37.1$\pm$3.7 |41.6$\pm$4.7 | 41.9$\pm$3.6 |**46.6$\pm$3.5** |
> |half-e | 35.2$\pm$2.3 | 40.0$\pm$4.5 | **63.7$\pm$5.9** | 55.1$\pm$6.6 |
> |hopper-m | 42.1$\pm$3.5 | **46.9$\pm$5.1** | 43.2$\pm$2.6 | 39.0$\pm$3.8 |
> |hopper-mr | 34.5$\pm$3.7 | 26.1$\pm$4.7 | 20.4$\pm$3.7 | **42.6$\pm$ 5.3** |
> |hopper-me | 2.5 $\pm$ 0.1 | 1.2$\pm$0.1 | **34.1$\pm$6.2** | **34.1$\pm$ 4.3** |
> |hopper-e | 43.1 $\pm$ 4.4 | 56.2$\pm$5.1 | **84.6$\pm$7.0** | 71.2$\pm$4.7 |
> | average | 31.9 | 34.3 | 42.6 | **43.8** |
>
>
> |Dataset (10\%)|IQL|IGDF|OTDF|DROCO|
> |-|-|-|-|-|
> |half-m | **45.2$\pm$0.1** | **45.2$\pm$0.1** |42.2$\pm$0.1 | **45.3$\pm$0.2** |
> |half-mr |22.1$\pm$2.8 | 22.9$\pm$1.4 |15.6$\pm$3.1 |**26.9$\pm$3.2** |
> |half-me |43.7$\pm$6.5 |57.1$\pm$8.9 | 46.7$\pm$4.4 |**60.1$\pm$7.1** |
> |half-e | 49.7$\pm$3.6 | 47.6$\pm$2.1 | **79.6$\pm$3.0** | 67.4$\pm$5.8 |
> |hopper-m | 48.8$\pm$2.1 | 54.3$\pm$6.6 | 46.3$\pm$3.7 | **55.4$\pm$5.3** |
> |hopper-mr | 40.2$\pm$5.7 | 30.0$\pm$5.2 | 26.2$\pm$4.4 | **47.3$\pm$ 7.0** |
> |hopper-me | 12.5 $\pm$ 0.4 | 11.6$\pm$0.6 | **58.1$\pm$4.9** | 54.0$\pm$ 6.4 |
> |hopper-e | 62.6 $\pm$ 6.9 | 70.1$\pm$3.2 | **97.0$\pm$3.3** | 89.3$\pm$9.6 |
> | average | 40.6 | 42.4 | 51.5 | **55.7** |
>
> |Dataset (20\%)|IQL|IGDF|OTDF|DROCO|
> |-|-|-|-|-|
> |half-m | 45.6$\pm$0.3 | 45.4$\pm$0.2 |43.7$\pm$0.1 | **47.8$\pm$0.1** |
> |half-mr |26.4$\pm$4.3 | 28.2$\pm$2.5 |21.9$\pm$2.7 |**32.8$\pm$4.5** |
> |half-me |50.6$\pm$4.8 |62.6$\pm$6.2 | 53.2$\pm$3.1 |**65.4$\pm$3.4** |
> |half-e | 57.7$\pm$5.5 | 60.5$\pm$4.8 | **83.4$\pm$1.8** | 74.6$\pm$3.9 |
> |hopper-m | 51.2$\pm$1.5 | 57.0$\pm$2.9 | 50.1$\pm$4.6 | **61.7$\pm$3.4** |
> |hopper-mr | **56.6$\pm$3.8** | 44.3$\pm$7.6 | 39.3$\pm$2.9 | 53.8$\pm$ 3.1 |
> |hopper-me | 29.2 $\pm$ 3.1 | 21.8$\pm$2.9 | 64.9$\pm$5.2 | **72.8$\pm$ 4.8** |
> |hopper-e | 76.5 $\pm$ 3.2 | 74.4$\pm$4.2 | **104.8$\pm$4.7** | 96.6$\pm$8.2 |
> | average | 49.2 | 49.3 | 57.7 | **63.2** |
>
> |Dataset (50\%)|IQL|IGDF|OTDF|DROCO|
> |-|-|-|-|-|
> |half-m | 49.2$\pm$0.1 | 48.5$\pm$0.3 |48.2$\pm$0.2 | **51.5$\pm$0.2** |
> |half-mr |34.7$\pm$3.8 | 39.2$\pm$4.2 |31.3$\pm$3.5 |**45.6$\pm$6.1** |
> |half-me |61.2$\pm$3.1 |70.3$\pm$4.7 | 61.4$\pm$2.6 |**77.6$\pm$5.0** |
> |half-e | 74.9$\pm$4.2 | 71.3$\pm$5.7 | **94.2$\pm$4.1** | 87.3$\pm$5.8 |
> |hopper-m | 67.7$\pm$4.9 | 75.5$\pm$4.6 | 62.8$\pm$3.8 | **80.6$\pm$6.2** |
> |hopper-mr | 69.9$\pm$4.5 | 61.5$\pm$2.7 | 59.2$\pm$4.3 | **74.2$\pm$ 4.3** |
> |hopper-me | 46.4 $\pm$ 2.5 | 49.4$\pm$5.3 | 73.8$\pm$3.7 | **81.3$\pm$ 2.5** |
> |hopper-e | 89.3 $\pm$ 4.5 | 86.1$\pm$3.6 | **111.1$\pm$3.2** | 103.4$\pm$3.9 |
> | average | 61.7 | 62.4 | 67.8 | **75.2** |
>
> The results show that the performance of all methods improves as the size of the target dataset increases, which is expected since the target data is crucial for policy learning. Furthermore, **DROCO consistently outperforms the baseline methods across all dataset sizes**. These results demonstrate the effectiveness and robustness of our method with varying amounts of target data.
>
> **When does RCB become too conservatism?**
>  Recall that the RCB operator includes an infimum operator in Equation (5). Therefore, if the ensemble dynamics model predicts a next-state $\hat{s}^\prime$ whose value $V(\hat{s}^\prime)$ is very small, then the RCB could be overly conservative due to Bellman backups. This situation is likely to occur when the target data is limited and the dynamics model is prone to overfitting, which is exactly **the reason why we further introduce the dynamic value penalty and Huber loss to tackle the underlying value estimation error**.

---

> ### Author Response · Authors · 2025-11-22
> **Reply to Reviewer z4fr (Part 5)**
>
> ## Concern 8: The source and target behavior policies are often different
>
> We agree that the source and target behavior policies may differ in practice. **Theoretically and practically, our method does not require them to be the same**. As noted in Lines 165–167, the assumption of identical behavior policies is made solely for notational simplicity​ and does not affect our derivations or conclusions. Specifically, Even if the behavior policies are different, such as $\hat{\\mu} _ 1(\\cdot)$ for source dataset, and $\\hat{\\mu} _ 2(\\cdot)$ for target dataset, we just need to replace $\\hat{\\mu}(\\cdot)$ to $\\hat{\\mu} _ 1(\\cdot)$ when $\\mathcal{M}=\\mathcal{M} _ {src}$ or $\\hat{\\mu} _ 2(\\cdot)$ when $\\mathcal{M}=\\mathcal{M} _ {tar}$. **All theoretical results remain valid** under this substitution.
>
> The reviewer may also be concerned about the empirical results when the source and target datasets share different behavior policies. We conduct additional experiments to address this concern. We consider four tasks (halfcheetah, hopper, walker2d, ant) with kinematic shifts. We relax the constraint of identical behavior policies, allowing the source and target datasets to have different qualities (medium or expert). For instance, a medium-quality source dataset may be paired with either a medium- or expert-quality target dataset. All other experimental settings follow Section 5.1, with IQL, IGDF, and OTDF as baselines. The results are presented below. We also incorporate the results in our Appendix E.7 in our revision.
>
> |Source|Target|IQL|IGDF|OTDF|DROCO|
> |-|-|-|-|-|-|
> |half-medium| medium | **45.2$\pm$0.1** | **45.2$\pm$0.1**  | 42.2$\pm$0.1 | **45.3$\pm$0.2** |
> |half-medium| expert | 47.5 $\pm$ 1.1 | 45.4$\pm$1.3| **58.3$\pm$ 2.8** | 52.6$\pm$4.2 |
> |half-expert| medium | 47.1$\pm$1.5  | 46.8$\pm$2.4  | 51.7$\pm$0.4  | **58.5$\pm$ 0.3** |
> |half-expert| expert | 49.7$\pm$3.6 | 47.6$\pm$2.1 | **79.6$\pm$3.0** |67.4$\pm$5.8 |
> |hopper-medium| medium | 48.8$\pm$2.1 | 54.3$\pm$6.6 | 46.3$\pm$3.7 | **55.4$\pm$5.3** |
> |hopper-medium| expert | 56.1$\pm$4.4 | 61.8$\pm$ 4.4 | 69.3$\pm$3.9 | **80.8$\pm$ 6.2** |
> |hopper-expert| medium | 53.6$\pm$2.4 | **61.3$\pm$4.7** | 51.4$\pm$2.1 | **62.2$\pm$4.6** |
> |hopper-expert| expert | 62.6 $\pm$ 6.9 | 70.1$\pm$3.2 | **97.0$\pm$3.3** | 89.3$\pm$9.6 |
> |walker2d-medium| medium | 48.7 $\pm$ 1.9 | 51.8$\pm$2.4 | 43.0$\pm$2.1 | **70.8$\pm$3.3** |
> |walker2d-medium| expert | 71.4$\pm$3.7 | 82.5$\pm$5.3 | 76.8$\pm$4.1 | **94.6$\pm$ 5.8** |
> |walker2d-expert| medium | 55.4$\pm$3.1 | 58.6$\pm$ 5.5 | 57.9$\pm$2.0 | **83.0$\pm$4.8** |
> |walker2d-expert| expert | 90.1 $\pm$ 3.2 | 93.7$\pm$5.8 | 98.9$\pm$2.1 | **106.0$\pm$0.8** |
> |ant-medium| medium | 89.9 $\pm$ 5.1 | 88.0$\pm$4.6 | 86.1$\pm$3.7 | **92.7$\pm$6.3** |
> |ant-medium| expert | 107.6$\pm$1.8 | **112.4$\pm$3.3** | 105.9$\pm$2.3 | 110.3$\pm$2.0 |
> |ant-expert| medium | 93.7$\pm$3.5 | 90.2$\pm$2.8 | 98.6$\pm$ 4.5| **100.4$\pm$ 2.3** |
> |ant-expert| expert | 111.0 $\pm$ 3.3 | **119.2$\pm$5.6** | 111.6$\pm$2.9 | **120.0$\pm$2.1** |
> |average| | 67.4 | 70.6 | 73.4 | **80.6** |
>
> **The results indicate that DROCO maintains its superiority over the baselines even when the source and target behavior policies differ**. It achieves the highest average score (80.6) and best performance on 12 out of 16 datasets. These findings demonstrate the effectiveness of DROCO in scenarios with differing behavior policies.
>
>
> We hope our responses above could address the reviewer's concerns. If there is still anything unclear, we are more than happy to have more discussions with the reviewer!
>
> [1] Towards Robust Offline RL Under Diverse Data Corruption. ICLR 2024
>
> [2] Survival Instinct in Offline Reinforcement Learning. NeurIPS 2023
>
> [3] Optimal Transport: Old and New.
>
> [4] Distributionally robust stochastic programming.
>
> [5] Robust reinforcement learning using offline data. NeurIPS 2022
>
> [6] Learning Cross-domain correspondence for control with dynamics cycle-consistency. ICLR 2021

---

> > ### Author Response · Authors · 2025-11-27
> >
> > Dear Reviewer z4fr,
> >
> > we deeply appreciate your valuable feedback to our work. We are reaching out to kindly check whether our responses have addressed your concerns. We are happy to address any additional concerns the reviewer might have.
> >
> > Best, Authors

---

### Meta-Review · Area_Chair_wrMD · 2026-01-06

**Summary:**

This paper studies “dual robustness” in cross-domain offline RL: robustness during training to source–target dynamics shift and robustness at deployment to perturbations around target dynamics. It proposes a Robust Cross-domain Bellman (RCB) operator that applies robust backups (min over an uncertainty set) on source data while using standard backups on target data, and implements it in DROCO with ensemble dynamics, a dynamic value penalty, and Huber loss. The key concerns shaping the decision were novelty/positioning (many theoretical pieces look standard from robust RL and the split-backup idea may be incremental), generality beyond pure dynamics shift (reward/observation/state-action shifts), practical tuning and conservatism (uncertainty set radius, sensitivity to hyperparameters, when it becomes overly conservative), and clarity around the value penalty/Huber components.

**Reviewer Concerns:**

The rebuttal addressed several major points: it clarified positioning as cross-domain dual robustness rather than new robust-RL theory, added experiments under observation shift and reward shift, explained extension limits beyond Wasserstein (TV via discrete metric; others require new dual forms), proposed an ensemble-based way to avoid manually tuning the uncertainty radius and reduce over-conservatism, expanded scaling studies with target-data size, and clarified value-penalty/Huber motivation. It also added experiments where source/target behavior policies differ and provided additional DMC-task results to support generalization, and one reviewer (k93k) explicitly kept their positive score. Remaining weaknesses are mainly around the “ICLR-level novelty” concern from z4fr (core idea still largely a split robust/standard backup with standard theory) and limited breadth of evaluation compared to broader cross-domain settings with heterogeneous state/action spaces (left as future work).

**Reviewer Scores:**

z4fr likely 4→5 (additional shift experiments, behavior-policy mismatch tests, target-data scaling, and clearer practical choices reduce several concerns, though novelty reservations remain). XtEu likely 6→6 (minor clarifications answered; already positive). k93k stays 6→6 (explicitly kept positive score). Overall sentiment remains mildly positive with one key novelty skeptic.

---

### Decision · Program_Chairs · 2026-01-26

Accept (Poster)